# MedAraBench: Large-Scale Arabic Medical Question Answering Dataset and Benchmark

**Mouath Abu-Daoud**[1]**, Leen Kharouf**[1]**, Omar El Hajj**[1]**, Dana El Samad**[1]**,
Mariam Al-Omari**[1]**, Jihad Mallat**[3]**, Khaled Saleh**[3]**, Nizar Habash**[2]**, Farah E.
Shamout**[1]

[1]Engineering Division, New York University Abu Dhabi, UAE
[2]Science Division, New York University Abu Dhabi, UAE
[3]Cleveland Clinic Abu Dhabi, UAE
{mma9138, lk2713, ose6432, dae7168, ma8058, nh48, fs999}@nyu.edu
{mallatj, salehk}@ccad.ae

## Abstract

Arabic remains one of the most underrepresented languages in natural language processing research, particularly in medical applications, due to the limited availability of open-source data and benchmarks. The lack of resources hinders efforts to evaluate and advance the multilingual capabilities of Large Language Models (LLMs). In this paper, we introduce MedAraBench, a large-scale dataset consisting of Arabic multiple-choice question-answer pairs across various medical specialties. We constructed the dataset by manually digitizing a large repository of academic materials created by medical professionals in the Arabic-speaking region. We then conducted extensive preprocessing and split the dataset into training and test sets to support future research efforts in the area. To assess the quality of the data, we adopted two frameworks, namely expert human evaluation and LLM-as-a-judge. Our dataset is diverse and of high quality, spanning 19 specialties and five difficulty levels. For benchmarking purposes, we assessed the performance of sixteen state-of-the-art open-source and proprietary models, such as GPT-5, Gemini 2.0 Flash, and Claude 4-Sonnet. Our findings highlight the need for further domain-specific enhancements. We also explore QLoRA fine-tuning on LLaMa-3.1-8B-instruct to assess our dataset's viability. We release the dataset and evaluation scripts to broaden the diversity of medical data benchmarks, expand the scope of evaluation suites for LLMs, and enhance the multilingual capabilities of models for deployment in clinical settings.

## 1 Introduction

The emergence of Large Language Models (LLMs) has driven transformative progress in Natural Language Processing (NLP) in recent years. They have demonstrated exceptional performance across various renowned benchmarks due to their powerful understanding and reasoning abilities, grounded in the vast amount of knowledge in their training corpora (Brown et al., 2020; Bommasani et al., 2022; Chowdhery et al., 2022). This includes general and domain-specific benchmarks (Wang et al., 2021; Stahlberg, 2020).

However, performance improvements remain variable across underrepresented languages and domains, particularly in high-stakes applications like medicine (Jiang et al., 2025; Yang et al., 2025). For example, Arabic is among the most spoken languages in the world, with over 400 million speakers across the globe. However, it remains underrepresented in the medical domain, mainly due to limited expert-annotated resources (Habash, 2005). Arabic NLP generally presents inherent language-specific linguistic challenges (Habash, 2005; Farghali & Shaalan, 2009; AlMoaiad, 2024), making the availability of such resources essential for

Table 1: Comparison of MedAraBench with existing medical QA benchmarks, including estimated dataset size.

| Benchmark | Language(s) | Type | Size | Expert Annotation | Difficulty Mapping | Specialty Coverage | Arabic | Public |
|---|---|---|---|---|---|---|---|---|
| MedQA | English, Chinese | MCQs | 60,000 | ✓ | × | ✓ | × | ✓ |
| MedMCQA | English | MCQs | 193,000 | ✓ | × | ✓ | × | ✓ |
| MMLU (USMLE) | English | MCQs | 1,800 | × | × | ✓ | × | ✓ |
| MMLU Translation | 14 incl. Arabic | MCQs | 15,000 | ✓ | × | ✓ | ✓ | ✓ |
| AraMed | Arabic | QA | 270,000 | ✓ | × | ✓ | ✓ | × |
| MedArabiQ | Arabic | QA and MCQs | 700 | × | × | ✓ | ✓ | ✓ |
| **MedAraBench (Ours)** | Arabic | MCQs | 24,000 | ✓ | ✓ | ✓ | ✓ | ✓ |

assessing the performance of LLMs, especially as they are being deployed in diverse medical contexts.

Several benchmarks have been recently introduced in the medical domain, as summarized in Table 1. However, most of them focus almost exclusively on English (Jin et al., 2021; 2019). Recent work began to address this need but remains limited in scope and size (Abu Daoud et al., 2025). Thus, there is a pressing need for large-scale benchmarks to assess and improve LLMs for Arabic-language medical reasoning.

To address those gaps, in this paper, we present MedAraBench, a comprehensive benchmark for evaluating and advancing LLMs on Arabic medical tasks. The dataset consists of curated Multiple-Choice Questions (MCQs) spanning different specialties and difficulty tiers aligned with stages of medical education. We propose a standardized development and evaluation protocol to enable reproducible and clinically meaningful assessment of LLMs. Our key contributions are as follows:

- We introduce MedAraBench, a large-scale Arabic medical benchmark featuring 24,883 MCQs across 19 medical specialties and five difficulty levels. The benchmark includes standardized training and test sets to enable systematic evaluation and advancement of LLMs.

- We perform extensive quality assessment via human expert evaluation, focusing on question clarity, clinical relevance, and medical correctness, as well as automated LLM-as-a-judge analysis.

- We benchmark 16 state-of-the-art proprietary and open-source LLMs across three categories (general purpose, Arabic-centric, medical) on the MedAraBench test set in the zero-shot setting to establish baseline performance for future research.

## 2 RELATED WORK

In recent years, several benchmark datasets have been developed to assess the capabilities of LLMs in the medical domain, driven by the expanding demand for applications that can streamline clinical workflows. Despite this progress, Arabic remains underrepresented in clinical NLP, mainly due to the lack of high-quality data to support building clinical applications in Arabic (Abdelaziz et al., 2025). As such, most existing benchmarks focus on English. For instance, the Massive Multitask Language Understanding (MMLU) benchmark includes question-answer pairs from the US Medical Licensing Exam (USMLE) (Hendrycks et al., 2021). Jin et al. (2020) introduce MedQA, a multilingual benchmark dataset consisting of multiple-choice questions sourced from medical licensing exams in English and Chinese. Pal et al. (2022) extend these benchmarks in MedMCQA to a multilingual evaluation framework but remains limited in Arabic.

Recent work has introduced new resources for medical evaluation in Arabic. Translations of existing datasets, such as of MMLU into 42 languages, including Arabic, provide valuable data but lack the necessary nuances for proper integration into clinical practice (Singh, 2025). AraMed presents an Arabic medical corpus and an annotated Arabic QA dataset sourced from online medical platforms (Alasmari et al., 2024). MedArabiQ presents one of

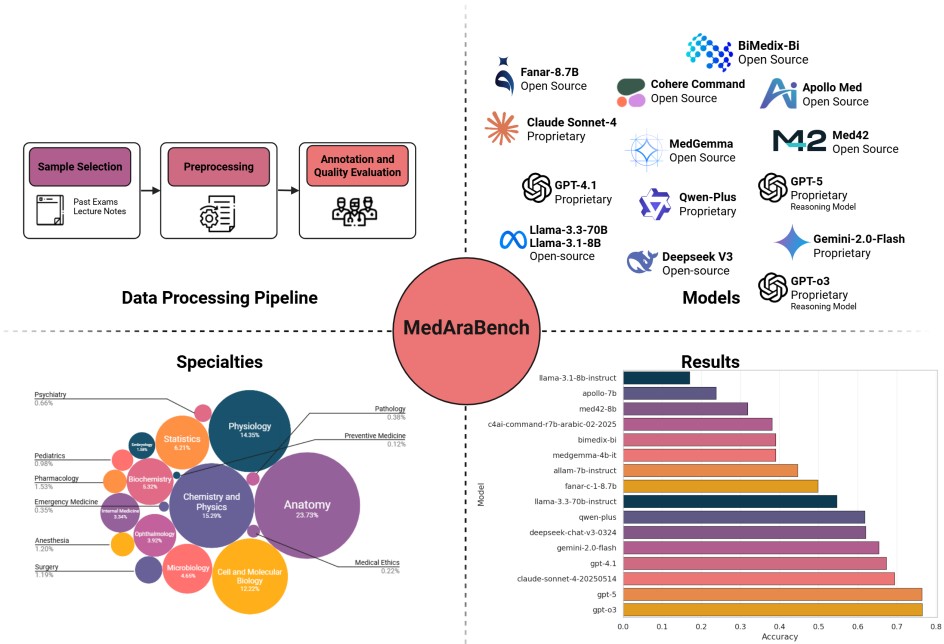

Figure 1: Overview of MedAraBench.

the first Arabic medical MCQ datasets, yet lacks specialty coverage, difficulty mapping, and expert evaluation (Abu Daoud et al., 2025).

Several evaluation frameworks have been proposed to evaluate the performance of clinical AI models. Kanithi et al. (2024) introduce 'MEDIC', a framework for evaluating LLMs from medical reasoning and ethics, to in-context learning and clinical safety. Wang et al. (2024) propose testing models on real-world input noise, dialogue interruptions, and reasoning justifications. Despite the growing interest in multilingual evaluation, there remains a critical gap in comprehensive, high-quality, and clinically relevant benchmarks for under-served languages. We aim to address this gap by introducing a comprehensive Arabic benchmark. Table 1 provides a structured comparison between MedAraBench and other existing medical benchmarks, covering key dimensions such as language coverage, specialty diversity, expert annotation, and public availability.

## 3 METHODOLOGY

Here, we describe the steps pertaining to data collection, processing, evaluation, benchmarking, few-shot learning, and finetuning, to facilitate proper reuse and fair comparison, aligning with best practices for benchmark construction and evaluation. An overview is provided in Figure 1.

### 3.1 DATA COLLECTION AND PRE-PROCESSING

We compiled a large repository of scanned paper-based exams, hosted on student-led social platforms of regional medical schools. The dataset did not include any personal or real patient data, so anonymization was not necessary, and our data collection complied with privacy and ethical standards. Considering the nature of the documents, we recruited professional typists to digitize the data. We then aggregated the paper-based exam documents to build a single MCQ dataset.

Upon manual inspection by NLP researchers, we observed that several documents exhibited issues such as missing or malformed correct answers, incomplete or duplicated answer choices, non-standard formatting or misaligned fields, and ambiguous answer keys or extra-

neous non-MCQ content. To ensure dataset quality and model compatibility, we applied strict filtering criteria to remove any questions with such issues. The filtering process was performed manually by five NLP researchers. While data acquisition and preprocessing required extensive effort, it highlights that the dataset is not publicly accessible in structured formats, thus reducing the likelihood of data contamination.

Each question is associated with several annotations: (i) number of answer choices (i.e., ABCD (4 choices), ABCDE (5 choices), and ABCDEF (6 choices)), (ii) difficulty level corresponding to five years of study (Y1 - Y5), and (iii) medical specialty. The questions fall under 19 medical specialties: Anatomy, Anesthesia, Biochemistry, Cell and Molecular Biology, Chemistry and Physics, Embryology, Emergency Medicine, Internal Medicine, Medical Ethics, Microbiology, Ophthalmology, Pathology, Pediatrics, Pharmacology, Physiology, Preventive Medicine, Psychiatry, Statistics, and Surgery. The scanned documents collected from the repositories were originally categorized by specialty, which allowed us to directly inherit the medical specialty classification for each question.

We started by omitting all questions with 6 answer choices due to the small size of this subset (9 questions), allowing us to categorize our data into two categories (4 answer choices and 5 answer choices). We then performed a stratified random split of the dataset into training (80%) and test sets (20%). This was to ensure that the medical specialties are represented evenly across the training and test sets (i.e., if the dataset contains 100 Cardiology questions, 80 would be randomly included in the training set, while the remaining 20 would be in the test set). A summary of the dataset in terms of token length distribution, medical specialty distribution, and difficulty level distribution is presented in Appendix A.

It is important to note that the homogeneity regarding medical terminology consistency is essential in assessing the capabilities of LLMs on standardized Arabic medical tasks. As it pertains to MedAraBench, we did not perform any external standardization across the dataset as we acknowledge that this may not be a major issue from a benchmarking perspective since real-world medical QA may not necessarily be standardized in terms of terminology.

We acknowledge that Arabic medical terminology standardization remains an ongoing challenge, with unified vocabulary frameworks still under development across Arabic-speaking regions (Amar, 2022), (Attia, 2024). The expert clinician evaluations described in Section 3.2 serve as quality control, with reviewers achieving high agreement on all metrics despite the absence of standardization.

## 3.2 Quality Assessment

To further assess the quality of our dataset, we conducted two analyses: human expert evaluation and using LLM-as-a-judge.

### 3.2.1 Human Expert Evaluation

We designed our expert evaluation protocol to assess the data according to the following criteria:

1. **Medical Accuracy:** the extent to which the question, options, and correct answer reflect current, evidence-based medical knowledge (Scale: high or low) (Olatunji et al., 2024; Iskander et al., 2024; Rejeleene et al., 2024).

2. **Clinical Relevance:** the practical importance and applicability of the question content to real-world medical practice or education (Scale: high or low) (Iskander et al., 2024; Olatunji et al., 2024).

3. **Question Difficulty:** the complexity required to answer the question correctly (Scale: high or low) (Iskander et al., 2024).

4. **Question Quality:** assessment of the MCQ construction quality (Scale: high or low) following established medical education standards (Al-Rukban, 2006):
   - *Clarity:* question is clear, complete, and unambiguous.

- *Option Homogeneity:* all distractors are plausible and of similar type.
- *Single Best Answer* one clearly correct option exists.
- *No Clueing:* options do not provide clues to other answers.

We selected samples for review from the test set and determined the sample size based on Cochran's formula (Cochran, 1977), to create a representative sample size (Hosseini, 2024). We estimated a single proportion at 95% confidence with a $\pm 5$ percentage-point margin of error, using $p = 0.5$ as a conservative assumption when the true quality rate is unknown because it maximizes variance and therefore yields a safe upper bound on sample size. We first calculated an estimated sample size assuming an infinite population using

$$n_0 = \frac{z^2 p(1-p)}{e^2}$$

with $z = 1.96$, $p = 0.5$, and $e = 0.05$. However, since the dataset is finite, we then applied the finite population correction using

$$n = \frac{n_0}{1 + \frac{n_0 - 1}{N}}$$

We recruited two board-certified clinicians specializing in Anesthesiology and Internal Medicine with Arabic clinical fluency and over 20 years of experience each. The reviews were double-blinded to model outputs and data provenance, and were conducted independently with pre-registered instructions on *Qualtrics* (Provo, UT, https://www.qualtrics.com), a web-based survey platform allowing for standardized presentation of evaluation criteria and independent response collection across both reviewers.

### 3.2.2 LLM-as-a-Judge

Considering that only a subset of the test set was considered for the human expert evaluation, we further introduced an LLM-as-a-judge evaluation protocol as an additional rating of data quality. Motivated by previous literature (Abu Daoud et al., 2025), we prompted four of our best-performing SOTA LLMs (gpt-o3, gemini-2.0-flash, and claude-4-sonnet) to act as medical education experts and to evaluate the MCQs along the same metrics used in our expert quality evaluation: Medical Accuracy, Clinical Relevance, Question Difficulty, and Question Quality, on a binary (0 or 1) scale for the entire test set. Additionally, we calculated Pearson Correlation coefficients for each model and the expert reviewers on the 378-question dataset evaluated by our medical experts. This approach gives us the advantage of providing a more nuanced evaluation across a broader set of our data, while also providing insights into the efficacy of LLMs in evaluating the quality of Arabic medical data relative to expert annotators.

### 3.3 Benchmarking Protocol

To introduce new benchmark results, we evaluated 16 proprietary and open-source models on the test set of the MedAraBench benchmark. We set the models' temperature as 0 to ensure stable outputs as shown in previous work for classification of MCQs (Abu Daoud et al., 2025). We selected one letter response per question.

- **Open-source models:** Llama-3.3-70b-instruct (Grattafiori et al., 2024a), Llama-3.1-8b-instruct (Grattafiori et al., 2024b), Deepseek-chat-v3-0324 (DeepSeek-AI, 2024), Allam-7b-instruct (Bari, 2024), Cohere c4ai-comman-r7b-arabic-02-2025 (Alnumay, 2025), Medgemma-4b-it (Sellergren, 2025), Apollo-7b (Wang, 2024), Fanar-C-1-8.7b (Abbas, 2025), BiMedix-Bi-27B (Pieri et al., 2024), and Med42-8b (Christophe, 2024).
- **Proprietary models:** Claude-sonnet-4-20250514 (Anthropic, 2025), Gemini-2.0-flash (Google Cloud, 2025), GPT-4.1 (OpenAI, 2025a), GPT-5 (OpenAI, 2025b), GPT-o3 (OpenAI, 2025c), and Qwen-plus (Yang & et al., 2024).

Our prompt is shown in Appendix B below. Answers were extracted via post-processing, whereas models were prompted to output the answer choice letter (A-D) directly, and we parsed their text responses using pattern matching to extract the answer. No set language parameters in API and model calls were used since models automatically detect Arabic text from the input. The results of our benchmarking experiments are shown in Table 4 below.

### 3.4 Few-shot Learning Protocol

Building upon our zero-shot benchmarking protocol, we conducted few-shot experiments on LLaMa-3.1-8B-instruct to assess in-context learning capabilities. We provided 3 high-quality sample questions that were rated highly across all evaluation metrics (question quality, clinical relevance, difficulty, and medical accuracy) by expert evaluators. These exemplar questions were carefully selected from the training split and omitted from the test set to ensure fair evaluation. The few-shot examples covered diverse medical topics including anatomy, biochemistry, and physiology, formatted with questions, options, and correct answers in Arabic to maintain linguistic consistency with the test items. The chosen questions and few-shot prompt are provided in Appendix B below.

### 3.5 Low-rank Adaptation

We further investigated parameter-efficient fine-tuning using QLoRA (Quantized Low-Rank Adaptation) on the Llama-3.1-8B-instruct model. The model was loaded in 4-bit precision and trained on the MedAraBench training split, formatted as Arabic prompt-response pairs for multiple-choice question answering. LoRA adapters were applied to key attention modules (q_proj, k_proj, v_proj, o_proj) with standard hyperparameters, training for up to 800 steps with batched gradient accumulation. This approach enabled efficient adaptation to Arabic medical terminology and reasoning patterns while preserving the models' general capabilities. The fine-tuned models were evaluated on the same test set used in our zero-shot and few-shot experiments to directly measure the impact of MedAraBench data on model performance.

## 4 Results

In this section, we present a summary of our dataset and the results of our expert quality evaluation, LLM-as-a-judge experiments, and benchmarking experiments.

### 4.1 Dataset Summary

The initial dataset consisted of 34,333 MCQs. The manual filtering process resulted in a reduction of approximately 29% of the initial dataset, yielding 24,883 samples overall. The training set consisted of 19,894 samples, and the test set of 4,989 samples. An overview of the dataset is shown in Figure A1 in Appendix A below, along with additional statistical summaries of the dataset.

### 4.2 Expert Quality Assessment

At 95% confidence and $\pm 5\%$ margin, Cochran's formula initially yielded $n_0 = 384$ questions. The final sample size was 378 after adjusting for a finite sample. Hence, our two annotators completed a review of 378 questions as a representative sample of the entire dataset. The results of our data quality assessment and the inter-annotator agreement are summarized in Table 2. Our results show slight to fair levels of agreement across all metrics, with Medical Accuracy having the highest level of agreement with a Cohen's Kappa score of 0.555 and a percentage agreement of 82%.

Additionally, we provide a detailed per-specialty breakdown of the evaluation results in Appendix C. To better assess the results, we investigate individual average and agreement scores for each specialty. Specifically, Figures C1, C2, C3, and C4 show the average accuracy per specialty for each metric, while Tables C1, C2, C3, and C4 show the average accuracy

Table 2: Expert quality assessment results for the representative data subset.

| Metric | Average [standard deviation] | Percent Agreement | Cohen's Kappa |
|---|---|---|---|
| Medical Accuracy | 0.722 [0.448] | 82.0% | 0.555 |
| Clinical Relevance | 0.653 [0.476] | 65.6% | 0.275 |
| Question Difficulty | 0.669 [0.471] | 65.6% | 0.233 |
| Question Quality | 0.767 [0.423] | 68.3% | 0.152 |

Table 3: Evaluation results of LLM-as-a-judge applied to the test set.

| Model | Evaluation Metric (average) | | | |
|---|---|---|---|---|
| | Medical Accuracy | Clinical Relevance | Question Difficulty | Question Quality |
| GPT-o3 | 0.673 [0.469] | 0.827 [0.378] | 0.588 [0.492] | 0.841 [0.366] |
| Gemini 2.0 Flash | 0.717 [0.450] | 0.565 [0.496] | 0.815 [0.388] | 0.774 [0.366] |
| Claude-4-Sonnet | 0.711 [0.453] | 0.749 [0.434] | 0.576 [0.494] | 0.764 [0.425] |
| GPT-5 | 0.533 [0.499] | 0.610 [0.488] | 0.597 [0.490] | 0.476 [0.499] |

and agreement results per specialty for each metric. All in all, our expert quality evaluations indicate that the dataset is of high quality with fair levels of agreement across a random sample of our test set.

Table 4: Benchmark accuracies, model sizes, and training dataset size for all evaluated LLMs.

| Model Type | Model Category | Model | Model Size | Training Dataset Size | Overall Accuracy |
|---|---|---|---|---|---|
| Proprietary | General-purpose | claude-sonnet-4-20250514 | Unknown | Unknown | 0.694 |
| | | gemini-2.0-flash | Unknown | Unknown | 0.654 |
| | | gpt-4.1 | Unknown | Unknown | 0.673 |
| | | gpt-5 | Unknown | Unknown | 0.764 |
| | | gpt-o3 | Unknown | Unknown | **0.765** |
| Open-source | General-purpose | deepseek-chat-v3-0324 | 8B parameters | 1.8 trillion | 0.620 |
| | | qwen-plus | 8B parameters | 18 trillion | 0.618 |
| | | llama-3.3-70b-instruct | 70B parameters | 15 trillion | 0.547 |
| | | llama-3.1-8b-instruct | 8B parameters | 15 trillion | 0.170 |
| | Arabic-centric | fanar-c-1-8.7b | 8.7B parameters | 1 trillion | 0.498 |
| | | allam-7b-instruct | 7B parameters | 5.2 trillion | 0.447 |
| | | c4ai-command-r7b-arabic-02-2025 | 7B parameters | Unknown | 0.381 |
| | Medical | medgemma-4b-it | 4B parameters | 4 trillion | 0.390 |
| | | apollo-7b | 7B parameters | Unknown | 0.238 |
| | | med42-8b | 8B parameters | 15T + 1B | 0.318 |
| | | bimedix-bi | 27B parameters | 632 million | 0.390 |

## 4.3 LLM-as-a-Judge Assessment

The results of our LLM-as-a-judge experiments are summarized in Table 3 and Table D1 in Appendix D across all four evaluation metrics. Our results show moderate agreement among different LLMs and comparable results with the expert evaluation scores. To better understand the results in comparison with expert evaluations, we provide a detailed breakdown of the LLM-as-a-judge results in Appendix D.

## 4.4 Benchmarking SOTA LLMs

We report all benchmarking results in Table 4. Our results show that general-purpose and reasoning-optimized models consistently outperform specialized or smaller open-source models on MedAraBench. GPT-o3 and GPT-5 both demonstrate top overall accuracy (0.765 and 0.764, respectively), while models like med42-8b and apollo-7b achieve the lowest scores (0.318 and 0.238, respectively). Tables E1 and E2 in Appendix E showcase a more detailed breakdown of the accuracy scores per specialty and level for each of the 15 evaluated models.

### 4.5 FEW-SHOT LEARNING AND QLoRA

The results of our few-shot learning and QLoRa protocols on Llama-3.1-8B-instruct are shown in Table 5 below.

Table 5: Few-shot, and QLoRA fine-tuning performance compared to baseline zero-shot accuracy for Llama-3.1-8B-instruct

| Model | Baseline Accuracy | Few-shot Accuracy | QLoRA Accuracy |
|---|---|---|---|
| llama-3.1-8b-instruct | 0.170 | 0.191 | 0.320 |

The few-shot and QLoRA fine-tuning experiments demonstrated substantial improvements over the baseline zero-shot performance. Few-shot learning provided a modest gain of 12.4% (from 0.170 to 0.191), while QLoRA fine-tuning, boosted accuracy by 88.2% to 0.320 - nearly doubling the model's performance.

## 5 DISCUSSION

Overall, in this study, we present MedAraBench, a new 25k question dataset consisting of both training and test sets. We report performance baseline results for SOTA LLMs and highlight critical differences in model capabilities under zero-shot settings. By exposing gaps in Arabic medical understanding, MedAraBench offers useful insights for the development of more inclusive, multilingual, and domain-specialized language models.

The expert quality assessment and LLM-as-a-judge experiments provide valuable insights into the quality of our data and the plausibility of using LLMs to evaluate the quality of medical datasets. Namely, the expert quality evaluation yields average scores ranging from 0.653 - 0.767 across all 4 evaluation metrics, with percent agreements and Cohen's Kappa scores ranging from 0.656 - 0.820 and 0.152 - 0.555, respectively, indicating slight to fair levels of agreement across all metrics. The average evaluation metric scores indicate moderate to high-quality data and fair agreement across annotators, but they indicate the need for the curation of more benchmark datasets of higher quality and clinical relevance to properly assess the readiness of LLMs for clinical deployment. Generally, we see weak to moderate alignment between LLM-as-a-judge evaluation and expert evaluation of the quality of our dataset on all metrics, except for question difficulty. Specifically, GPT-o3 has the highest Pearson Correlation scores with both experts A and B across Medical Accuracy (0.577 and 0.505, respectively), Clinical Relevance (0.252 and 0.377, respectively), and Question Quality (0.407 and 0.336, respectively). As for Question Difficulty, we see weak to no alignment across all models, with Gemini-2.0-Flash scoring highest with Expert A (0.019) and Claude-4-Sonnet scoring higher with Expert B (0.039). Our results indicate that LLMs are yet to be reliable for proper evaluation of Arabic medical data, and highlight the need for further training to better align with expert human standards.

Our benchmark evaluation results coincide with prior research on LLM benchmarking in the medical domain, whereas proprietary models typically outperform open-access models in structured tasks such as multiple-choice QA. This was previously demonstrated by Chen et al. (2025) and Alonso et al. (2024), who demonstrated superior accuracy performance by proprietary models relative to open-source models in medical QA tasks across multiple languages. This was further shown by Abu Daoud et al. (2025), who demonstrated superior performance by proprietary models such as Gemini 1.5 Pro, Claude 3.5 Sonnet, and GPT-4 in Arabic medical MCQ. Our results reinforce those findings, with all proprietary models performing at significantly higher or similar accuracy scores to open-source models. Namely, the proprietary models GPT-o3 and GPT-5 achieved the highest overall accuracies on MedAraBench (0.765 and 0.764, respectively), with Claude-Sonnet-4 also performing strongly (0.694). By contrast, even the largest open-source general-purpose models such as llama-3.3-70b-instruct and qwen-plus only reached 0.547 and 0.618, while domain-focused Arabic-centric and medical models, such as allam-7b-instruct (0.447), fanar-c-1-8.7b (0.498), and medgemma-4b-it (0.390), remained below 0.5 accuracy. While training dataset size and

model scale correlate with performance (e.g., larger models like llama-3.3-70b-instruct and deepseek-chat-v3-0324 outperformed smaller Arabic-centric or medical-only models), this was not sufficient for open-source models to match proprietary LLMs. We also observe that general-purpose models fine-tuned for reasoning consistently outperform Arabic-focused or medical-specific models, underscoring the importance of both architectural and data scale, as well as transfer learning capabilities honed in proprietary labs.

The superior performance by proprietary models is possibly due to the larger training corpora, stronger pretraining on structured datasets, more extensive instruction tuning, and specialized reinforcement learning pipelines that proprietary models undergo relative to their open-source counterparts. Additionally, our results show significantly higher performance by reasoning models (gpt-5 and gpt-o3) relative to other models, showing the promise and importance of incorporating reasoning and explainability into medical NLP as a whole and Arabic medical NLP specifically.

The few-shot learning and QLoRA experiments in Table 5 above revealed substantial improvements over the baseline zero-shot performance for Llama-3.1-8B-instruct, though with notable differences in effectiveness between the two approaches. Few-shot learning provided a modest gain of 12.4% (from 0.170 to 0.191), suggesting that in-context examples helped the model better understand the medical question format and reasoning patterns. Additionally, QLoRA fine-tuning, boosted accuracy by 88.2% to 0.320, nearly doubling the model's performance. The dramatic superiority of QLoRA fine-tuning highlights that parameter-efficient adaptation using domain-specific medical data is substantially more effective than in-context learning alone for adapting general-purpose models to specialized medical QA tasks in Arabic. These results emphasize the importance of targeted training protocols over prompt engineering for achieving competent performance in specialized medical language understanding tasks.

We further evaluate the evolution of models across generations by comparing contemporary and legacy model performance on the MedArabiQ benchmark (Abu Daoud et al., 2025). Our results, presented in Table G1 in Appendix G below, show that all contemporary models outperform legacy models on the MCQ task. Additionally, this analysis allows us to compare the merit of both the MedArabiQ and MedAraBench benchmarks, with gemini-2.0-flash, gpt-4.1, gpt-5, gpt-o3, and qwen-plus performing better on the MedArabiQ benchmark, while claude-sonnet-4-20250514 and llama-3.3-70b-instruct perform better on MedAraBench. This indicates that the MedAraBench benchmark is more challenging overall, highlighting its value in advancing medical Arabic NLP.

However, despite the significant improvement across generations, the highest performing model achieved an accuracy score of 0.765, which does not match expert-level performance and indicates clear headroom for improvement before being ready for deployment in clinical settings. Furthermore, this allows us to raise important questions about what exactly LLMs are learning. High accuracy does not necessarily indicate deep understanding or clinical reasoning. Instead, models may be leveraging statistical associations and lexical patterns to eliminate implausible answers. For example, frequent exposure to certain disease-treatment pairs during pretraining may allow models to make educated guesses without reasoning through symptom progression or differential diagnosis. This distinction is crucial, particularly in high-stakes applications such as medical education or decision support. Future work should consider evaluating not just answer correctness, but also the rationale behind model choices, possibly through explanation-based tasks or clinician scoring of model justifications.

## 6 LIMITATIONS AND FUTURE WORK

While our study provides a comprehensive evaluation of LLMs on Arabic medical MCQs and represents a substantial advancement in benchmarking capabilities, several limitations exist. First, the dataset is limited in its capability to evaluate LLMs on classification tasks only due to the nature of the MCQ dataset, preventing the evaluation of LLMs in generative tasks. Additionally, although the data source was not available in a structured digital format and required extensive digitization and cleaning efforts, we cannot certainly rule out contamination. Furthermore, our data assumes fluency in Modern Standard Arabic, which,

despite being common in formal settings, may not fully align with linguistic realities. This can affect the generalizability of MedAraBench to learners or practitioners accustomed to dialectal or mixed-language instruction.

Another limitation emerges from our expert quality evaluation experiments. Although our dataset was reviewed by two expert clinicians, we observed occasional inconsistencies across their assessments. We acknowledge that expert disagreement and inherent subjectivity are common in clinical judgment, but recognize the need for broader consensus in future validation efforts. Furthermore, our data is largely skewed toward easier difficulty levels (Y1 at 56% and Y2 at 22.89%) relative to more advanced levels (Y3 at 12.04%, Y4 at 3.38%, and Y5 at 5.14%. This distribution is due to the availability of source materials collected from regional medical repositories, which contained disproportionately more early-year content. While this evaluation limits the benchmark's capacity to assess advanced clinical reasoning, it is representative of the present educational resource landscape in Arabic, and provides value by establishing a baseline performance for Arabic medical understanding across the present difficulty levels and specialties, revealing that even basic medical reasoning remains challenging for current LLMs. Moreover, our data is text-only in its format, limiting its applicability to accommodating image-based reasoning required in specialties such as radiology and dermatology, and warranting expansions to other data modalities in future work.

In this study, we primarily focused on zero-shot evaluation of model performance, providing an assessment without further adaptation. While this is an important evaluation framework, future work could explore the impact of few-shot and chain-of-thought prompting strategies, as well as fine-tuning as an opportunity to improve model performance. Furthermore, future work could warrant the incorporation of dialectal data to enhance model adaptability across diverse Arabic clinical settings. Another important area of future work is to investigate the use of Arabic-based lecture notes to advance medical Arabic NLP beyond MCQs.

## 7 Conclusions

To conclude, we introduced a large-scale Arabic medical benchmark designed to evaluate the zero-shot performance of LLMs on curated MCQs. Covering 19 medical specialties and spanning five difficulty levels, MedAraBench provides a comprehensive and fine-grained lens for assessing Arabic medical reasoning in LLMs. Our benchmark provides an advancement for developing benchmarks in the Arabic language and exposes limitations in the performance of current LLMs in low-resource language tasks and the need for robust multilingual training strategies. Future work should explore fine-tuning strategies and the curation of larger and higher-quality datasets tailored to Arabic medical contexts. We release MedAraBench in hopes of supporting downstream clinical applications, and we hope that it serves as a catalyst for continued research at the intersection of Arabic NLP and medical AI.

### Ethics Statement

The authors disclose the use of generative AI tools to assist with LaTeX code cleanup and formatting only, with all content, analyses, and conclusions authored and verified by the researchers involved in this project.

### Reproducibility Statement

The MedAraBench dataset is available at `https://github.com/nyuad-cai/MedAraBench`

### Acknowledgements

This work was supported by the Meem Foundation, the NYUAD Center for Artificial Intelligence and Robotics, funded by Tamkeen under the NYUAD Research Institute Award CG010, the Center for Cyber Security (CCS), funded by Tamkeen under the NYUAD Research Institute Award G1104, and the Center for Interdisciplinary Data Science & AI (CIDSAI), funded by Tamkeen under the NYUAD Research Institute Award CG016. The research was carried out on the High Performance Computing resources at New York University Abu Dhabi (Jubail).

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

# A  DATA ANALYSIS

## A.1  DATA OVERVIEW

An overview of the MedAraBench dataset according to the distribution of questions per difficulty level and specialty is shown in Figure A1 below.

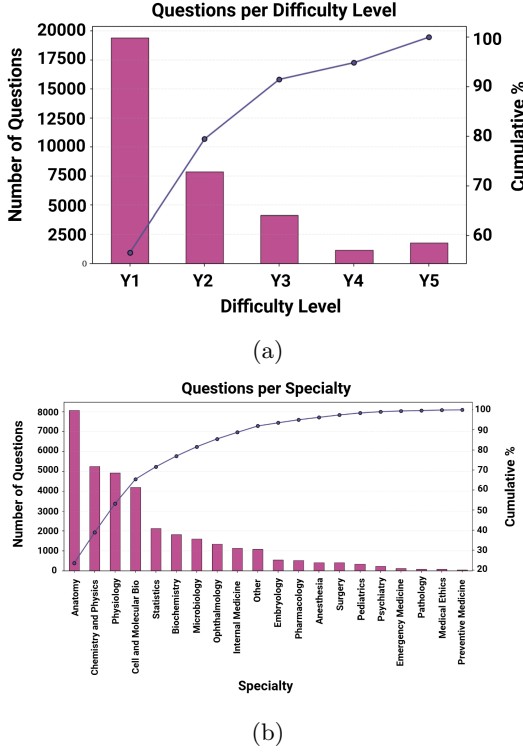

(a)

(b)

Figure A1: Overview of dataset according to difficulty level and specialties.

In terms of specialty, the largest subsets of the dataset falls under Anatomy (6100 questions - 24.61% of the dataset) and Physiology (3302 - 13.32%), while the smallest subset falls under Pathology (56 - 0.23%). A more detailed breakdown of the distribution of questions according to specialty can be shown in Table A1 and Figure A1 (b) above.

On the other hand, in terms of difficulty level, the largest subset of the dataset falls under Y1 (15095 questions - 60.89% of the dataset) while the smallest subset falls under Y4 (1033 - 4.17%). A more detailed breakdown of the distribution of questions according to level can be shown in Table A2 and Figure A1 (a) above. Additionally, Figure A2 shows the the composition of the five levels across the 16 specialties included in MedAraBench.

Table A1: Distribution of questions per medical specialty.

| Medical Specialty | Number of Questions | Percentage |
|---|---|---|
| Anatomy | 6100 | 24.61% |
| Anesthesia | 405 | 1.63% |
| Biochemistry | 1387 | 5.59% |
| Cell and Molecular Biology | 3155 | 12.73% |
| Chemistry and Physics | 3894 | 15.71% |
| Embryology | 184 | 0.74% |
| Emergency Medicine | 120 | 0.48% |
| Internal Medicine | 762 | 3.07% |
| Microbiology | 750 | 3.03% |
| Ophthalmology | 1318 | 5.32% |
| Pathology | 56 | 0.23% |
| Pharmacology | 329 | 1.33% |
| Physiology | 3302 | 13.32% |
| Statistics | 1795 | 7.24% |
| Surgery | 357 | 1.44% |

Table A2: Distribution of questions per difficulty level.

| Difficulty Level | Number of Questions | Percentage |
|---|---|---|
| Y1 | 15095 | 60.89% |
| Y2 | 4954 | 19.98% |
| Y3 | 2313 | 9.33% |
| Y4 | 1033 | 4.17% |
| Y5 | 1396 | 5.63% |

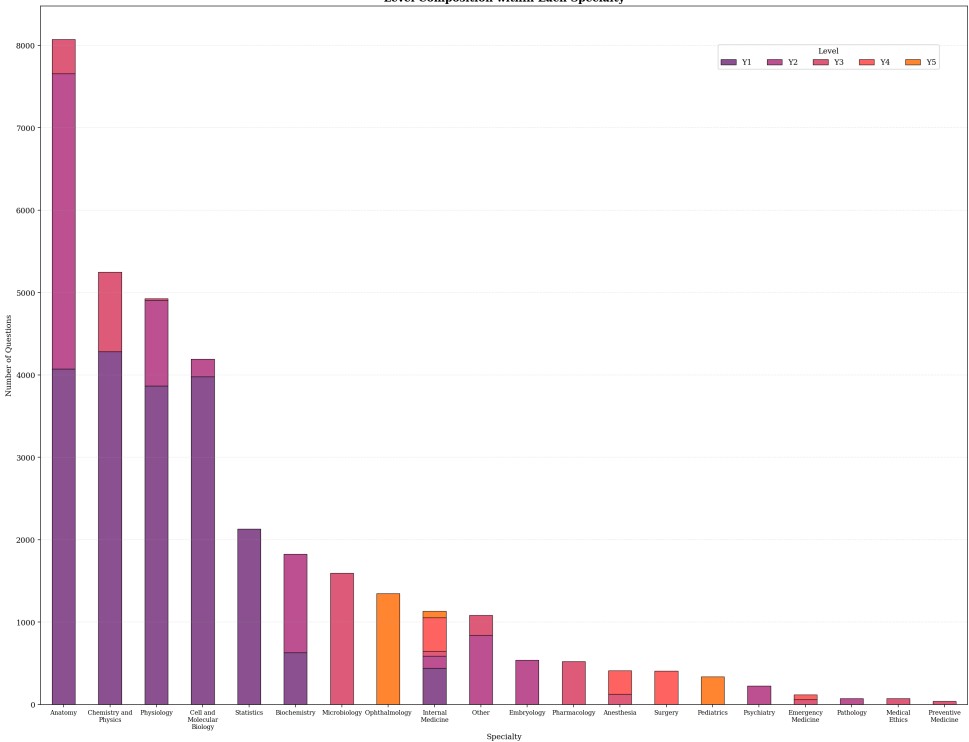

Figure A2: Level composition of the 5 different difficulty levels (Y1 - Y5) within each specialty in the MedAraBench dataset.

## A.2 TOKEN LENGTH DISTRIBUTION

There are 24,791 questions in our dataset. The questions are moderate in length, with a total average length of 37.86 characters. The answers vary in format, whereas 18554 questions have four answer choices (A, B, C, and D) and 6228 questions have five answer choices (A, B, C, D, and E). The average answer length across the datasets is 171.09 characters. Figure A3 below gives a detailed breakdown of the distributions of text lengths of the entire dataset.

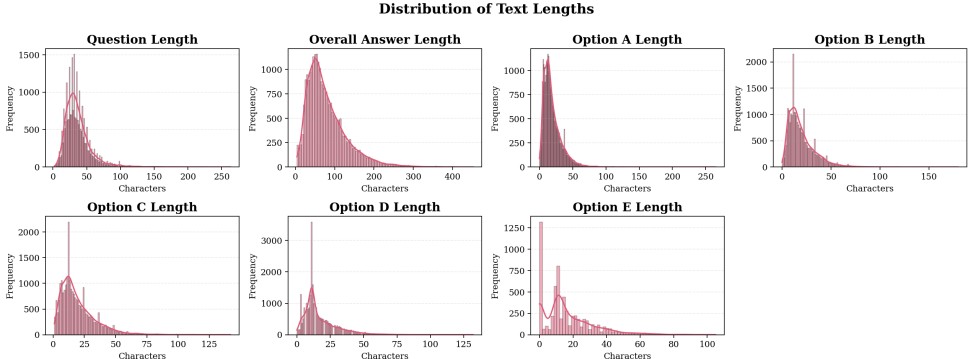

Figure A3: Distribution of text lengths in the MedAraBench dataset: (a) Distribution of question length; (b) Distribution of answer length; (c) Distribution of Option A length; (d) Distribution of Option B length; (e) Distribution of Option C length; (f) Distribution of Option D length; and (g) Distribution of Option E length.

## B  BENCHMARKING AND FEW-SHOT LEARNING PROMPTS

### B.1  BENCHMARKING PROMPT

Listing 1: Benchmarking prompt used for medical MCQ evaluation

```
"You are an expert medical virtual assistant.
Please provide the correct answer letter (A, B, C, or D)
for the following Arabic medical multiple-choice question.
Question:
{question_text_in_Arabic}

Options:
A:{option_A_in_Arabic}
B:{option_B_in_Arabic}
C:{option_C_in_Arabic}
D:{option_D_in_Arabic}
Answer:"
```

### B.2  FEW-SHOT LEARNING PROMPT

Listing 2: Few-shot prompt for medical MCQ evaluation

```
"You are an expert medical assistant. You will be provided with
    a few Arabic
medical sample questions and their correct answers, then a new
   question
that you must answer.

Sample Questions:
{sample_questions}

Now, the new question:
Question: {question}
Options:
{options_text}
(Please answer with the letter choice {allowed_letters} ONLY):"
```

The sample questions are as follows:

السؤال: المفصل الحلقي الطرجهالي من المفاصل؟
الخيارات: أ: الليفي, ب: الزليلي, ج: الزجاجي, د: المرن
الإجابة: ب

السؤال: فيما يخص العصب الزندي كل مما يلي صحيح ماعدا
الخيارات: أ: ينشأ من الحبل الأنسي للضفيرة العضدية, ب: يدخل الساعد بين رأسي الكبة المدورة, ج: يخترق الحاجز الأنسي
للعضد برفقة الشريان الجانبي الزندي العلوي, د: يكون وحشي وتر قابضة الرسغ الزندية عند المعصم, هـ: يعصب العضلة قابضة
الرسغ الزندية
الإجابة: ب

السؤال: من أسباب نقص سكر الدم الارتكاسي:
الخيارات: أ: نقص سكر الدم الهضمي, ب: الحساسية للوسين, ج: الغالاكتوزيمية, د: كل ما سبق صحيح
الإجابة: د

## C    EXPERT QUALITY ASSESSMENT DETAILS

In this appendix, we provide a detailed breakdown of the expert evaluation results introduced in Section 2. While the main text summarizes overall averages and agreement levels across all specialties, here we report per-specialty results to give a more fine-grained view of model performance and annotator consistency.

Figures C1–C4 present the distribution of annotator scores across specialties for each of the four evaluation metrics (Medical Accuracy, Clinical Relevance, Question Difficulty, and Question Quality). The corresponding Tables C1– C4 report the average scores, number of evaluated questions, percentage agreement, and Cohen's Kappa values for each specialty. Together, these results highlight the variability across domains and provide context for interpreting the aggregate quality metrics shown in the main text.

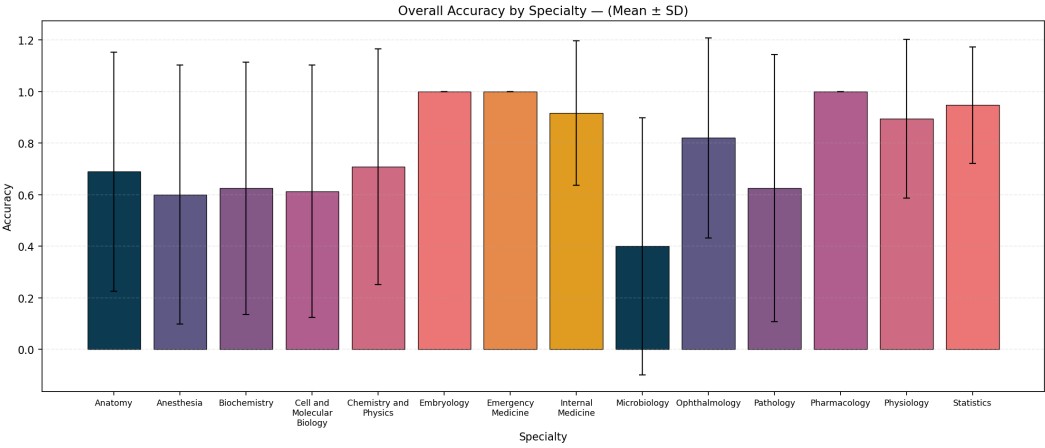

Figure C1: Overall annotator accuracy scores per specialty.

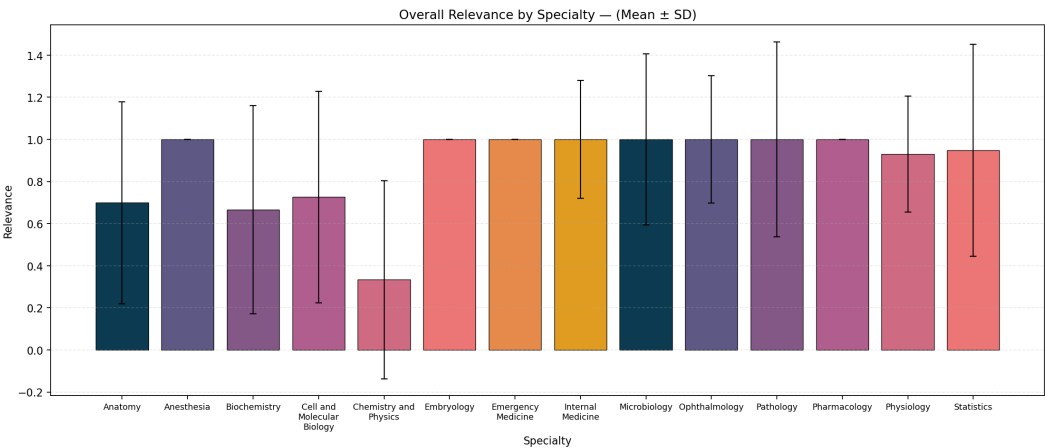

Figure C2: Overall annotator relevance scores per specialty.

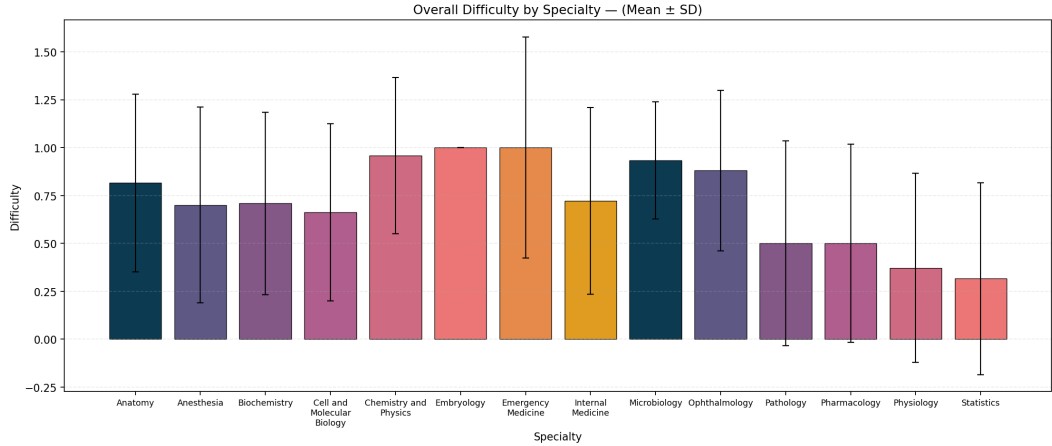

Figure C3: Overall annotator difficulty scores per specialty.

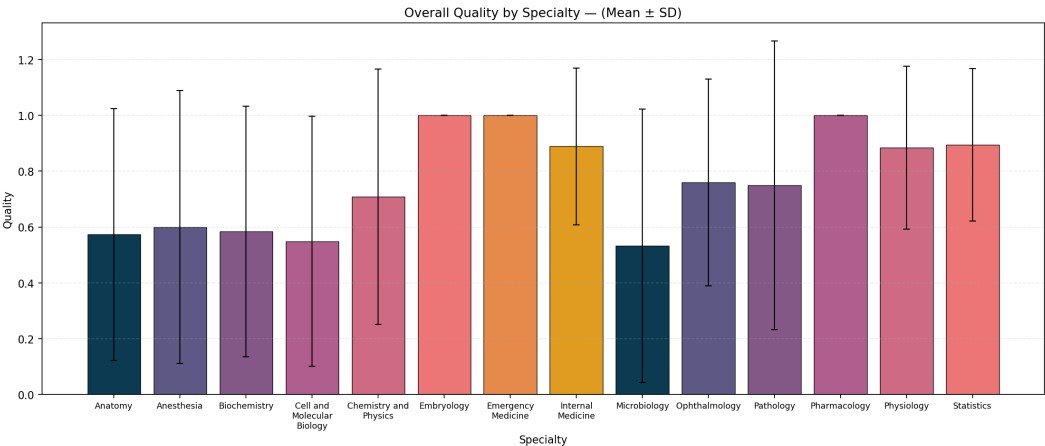

Figure C4: Overall annotator quality scores per specialty.

Table C1: Annotator accuracy scores per specialty.

| Specialty | Average [standard deviation] | Number of Questions | Percent Agreement | Cohen's Kappa |
|---|---|---|---|---|
| Anatomy | 0.689 [0.464] | 103 | 0.748 | 0.431 |
| Anesthesia | 0.600 [0.503] | 10 | 0.600 | 0.167 |
| Biochemistry | 0.625 [0.489] | 24 | 0.750 | 0.471 |
| Cell and Molecular Biology | 0.613 [0.489] | 62 | 0.774 | 0.529 |
| Chemistry and Physics | 0.708 [0.457] | 48 | 0.917 | 0.798 |
| Embryology | 1.000 [0.000] | 1 | 1.000 | - |
| Emergency Medicine | 1.000 [0.000] | 2 | 1.000 | - |
| Internal Medicine | 0.917 [0.280] | 18 | 0.944 | 0.640 |
| Microbiology | 0.400 [0.498] | 15 | 0.733 | 0.455 |
| Ophthalmology | 0.820 [0.388] | 25 | 0.880 | 0.603 |
| Pathology | 0.625 [0.518] | 4 | 0.750 | - |
| Pharmacology | 1.000 [0.000] | 4 | 1.000 | - |
| Physiology | 0.895 [0.308] | 43 | 0.884 | 0.380 |
| Statistics | 0.947 [0.226] | 19 | 1.000 | 1.000 |

Table C2: Annotator relevance scores per specialty.

| Specialty | Average [standard deviation] | Number of Questions | Percent Agreement | Cohen's Kappa |
|---|---|---|---|---|
| Anatomy | 0.699 [0.479] | 103 | 0.641 | 0.225 |
| Anesthesia | 1.000 [0.000] | 10 | 1.000 | - |
| Biochemistry | 0.667 [0.494] | 24 | 0.708 | 0.400 |
| Cell and Molecular Biology | 0.726 [0.501] | 62 | 0.435 | 0.095 |
| Chemistry and Physics | 0.333 [0.470] | 48 | 0.688 | 0.286 |
| Embryology | 1.000 [0.000] | 1 | 1.000 | - |
| Emergency Medicine | 1.000 [0.000] | 2 | 1.000 | - |
| Internal Medicine | 1.000 [0.280] | 18 | 0.833 | 0.000 |
| Microbiology | 1.000 [0.407] | 15 | 0.600 | 0.000 |
| Ophthalmology | 1.000 [0.303] | 25 | 0.800 | 0.000 |
| Pathology | 1.000 [0.463] | 4 | 0.500 | - |
| Pharmacology | 1.000 [0.000] | 4 | 1.000 | - |
| Physiology | 0.930 [0.275] | 43 | 0.884 | 0.224 |
| Statistics | 0.947 [0.504] | 19 | 0.211 | 0.021 |

Table C3: Annotator difficulty scores per specialty.

| Specialty | Average [standard deviation] | Number of Questions | Percent Agreement | Cohen's Kappa |
|---|---|---|---|---|
| Anatomy | 0.816 [0.464] | 103 | 0.650 | 0.240 |
| Anesthesia | 0.700 [0.510] | 10 | 0.500 | 0.074 |
| Biochemistry | 0.708 [0.476] | 24 | 0.750 | 0.442 |
| Cell and Molecular Biology | 0.661 [0.463] | 62 | 0.710 | 0.320 |
| Chemistry and Physics | 0.958 [0.408] | 48 | 0.625 | 0.027 |
| Embryology | 1.000 [0.000] | 1 | 1.000 | - |
| Emergency Medicine | 1.000 [0.577] | 2 | 0.000 | - |
| Internal Medicine | 0.722 [0.487] | 18 | 0.611 | 0.182 |
| Microbiology | 0.933 [0.305] | 15 | 0.800 | -0.098 |
| Ophthalmology | 0.880 [0.418] | 25 | 0.720 | 0.229 |
| Pathology | 0.500 [0.535] | 4 | 1.000 | - |
| Pharmacology | 0.500 [0.518] | 4 | 0.750 | - |
| Physiology | 0.372 [0.494] | 43 | 0.651 | 0.281 |
| Statistics | 0.316 [0.500] | 19 | 0.368 | -0.009 |

Table C4: Annotator quality scores per specialty.

| Specialty | Average [standard deviation] | Number of Questions | Percent Agreement | Cohen's Kappa |
|---|---|---|---|---|
| Anatomy | 0.573 [0.451] | 103 | 0.573 | 0.044 |
| Anesthesia | 0.600 [0.489] | 10 | 0.700 | 0.348 |
| Biochemistry | 0.583 [0.449] | 24 | 0.625 | 0.143 |
| Cell and Molecular Biology | 0.548 [0.448] | 62 | 0.484 | -0.120 |
| Chemistry and Physics | 0.708 [0.457] | 48 | 0.792 | 0.496 |
| Embryology | 1.000 [0.000] | 1 | 1.000 | - |
| Emergency Medicine | 1.000 [0.000] | 2 | 1.000 | - |
| Internal Medicine | 0.889 [0.280] | 18 | 0.944 | 0.640 |
| Microbiology | 0.533 [0.490] | 15 | 0.533 | 0.037 |
| Ophthalmology | 0.760 [0.370] | 25 | 0.840 | 0.432 |
| Pathology | 0.750 [0.518] | 4 | 0.750 | - |
| Pharmacology | 1.000 [0.000] | 4 | 1.000 | - |
| Physiology | 0.884 [0.292] | 43 | 0.860 | 0.178 |
| Statistics | 0.895 [0.273] | 19 | 0.842 | -0.075 |

# D LLM-as-a-Judge Assessment

The models were provided with the full test set of MCQ (stem, options, and correct answer) and instructed to return only valid JSON output. The exact prompt was:

Listing 3: Prompt provided to LLMs

```
You are a medical education expert. Evaluate the following
    multiple-choice
question (MCQ) on a binary scale (0 or 1) for each of the
    following metrics:

1. Medical Accuracy (1=high, 0=low)
2. Clinical Relevance (1=high, 0=low)
3. Question Difficulty (1=high, 0=low)
4. Question Quality (1=high, 0=low)
Important: Return ONLY valid JSON. No explanations, no markdown
    , no text.
The response must be exactly like this:

{
  "medical_accuracy": <0-1>,
  "clinical_relevance": <0-1>,
  "question_difficulty": <0-1>,
  "question_quality": <0-1>
}

Question stem: {row['Question']}
Options:
{options_text}
Correct answer: {row['Correct Answer']}
```

This setup ensured that LLM outputs were standardized and machine-readable. By aggregating scores across thousands of test questions, we obtained descriptive statistics and model-wise distributions that enabled a more fine-grained analysis than binary human ratings alone.

To examine agreement between models and expert evaluators, Pearson correlation coefficients were calculated on a per-question basis. The results of this analysis are shown in Table D1 below. Our Pearson Correlation results show GPT-o3 was the best performing model in terms of alignment with expert evaluations. As such, we opted to use GPT-o3 to complete LLM-as-aJudge evaluations over the entire training set of MedAraBench, allowing us to have a full evaluation of the entire dataset. Additionally, we ran t-tests to compare GPT-o3 scores on the training and test sets and found no significant difference on all 4 evaluation metrics, indicating consistent quality across splits. The results of this experiment are shown in table D2 below.

Table D1: Pearson correlation coefficients between model and expert ratings.

| Model | Expert A | | | | Expert B | | | |
|---|---|---|---|---|---|---|---|---|
| | GPT-o3 | Claude-4-Sonnet | Gemini-2.0-Flash | GPT-5 | GPT-o3 | Claude-4-Sonnet | Gemini-2.0-Flash | GPT-5 |
| Medical Accuracy | **0.577** | 0.065 | 0.023 | 0.043 | **0.505** | 0.053 | 0.053 | 0.068 |
| Clinical Relevance | **0.252** | 0.165 | 0.176 | 0.071 | **0.377** | 0.131 | 0.131 | 0.104 |
| Question Quality | **0.407** | 0.007 | 0.023 | 0.044 | **0.336** | 0.062 | 0.062 | 0.033 |
| Question Difficulty | -0.114 | -0.070 | **0.019** | -0.116 | -0.018 | **0.039** | **0.039** | -0.049 |

Table D2: GPT-o3 LLM-as-a-judge scores on training and test sets (mean [std]) with $t$-test p-values.

| Dataset | Medical Accuracy | Clinical Relevance | Question Quality | Question Difficulty |
|---|---|---|---|---|
| Training set | 0.638 [0.481] | 0.821 [0.383] | 0.839 [0.367] | 0.561 [0.496] |
| Test set | 0.673 [0.469] | 0.827 [0.378] | 0.588 [0.492] | 0.841 [0.366] |
| P-value | 0.0001 | 0.0362 | 0.0001 | 0.0001 |

# E    PERFORMANCE PER SPECIALTY AND LEVEL

To complement the overall evaluation, we analyze model accuracy across different medical specialties and difficulty levels. This breakdown highlights domain-specific strengths and weaknesses, as well as how performance varies across the five curriculum levels (Y1–Y5). We provide a detailed breakdown of model accuracies per specialty and levels in Tables E1 and E2 below.

In terms of specialty performance, our analysis reveals clear superior performance across models, with GPT-5 and GPT-o3 dominating most domains while apollo-7b and med42-8b consistently underperformed. Notably, the smaller medgemma-4b-it excelled in Pathology (0.574), demonstrating that specialized training can sometimes overcome scale limitations.

As for difficulty level performance, all models exhibited an accuracy drop from Y1 to Y3 before recovering in Y4-Y5. GPT-5 maintained the highest performance across four of five levels, while weaker models showed steeper declines at intermediate levels. This consistent pattern indicates that Y3 questions, which likely require more complex clinical reasoning, present the greatest challenge across all model architectures.

The significant performance variation between difficulty levels highlights critical limitations in current LLMs for medical applications. The average 15-20% performance drop from Y1 to Y3 suggests that while models excel at factual recall in early curriculum years, they struggle with the integrative clinical reasoning required at intermediate levels. This difficulty scaling effect was most pronounced for smaller models, indicating that scale and sophisticated training are crucial for handling complex medical reasoning tasks. These findings suggest LLMs may be better suited for foundational knowledge and specialized domains than for complex clinical decision-making.

Table E1: Overall Model Accuracy per Specialty.

| Model | Anatomy | Anesthesia | Biochemistry | Embryology | Emergency Medicine | Internal Medicine | Microbiology | Ophthalmology | Other | Pathology | Pharmacology | Physiology | Statistics | Surgery |
|---|---|---|---|---|---|---|---|---|---|---|---|---|---|---|
| allam-7b-instruct | 0.402 | 0.469 | 0.458 | 0.333 | 0.478 | 0.351 | 0.338 | 0.320 | 0.151 | 0.463 | 0.409 | 0.484 | 0.624 | 0.430 |
| apollo-7b | 0.315 | 0.198 | 0.237 | 0.000 | 0.435 | 0.175 | 0.192 | 0.178 | 0.076 | 0.259 | 0.106 | 0.183 | 0.253 | 0.109 |
| c4ai-command-r7b-arabic-02-2025 | 0.322 | 0.173 | 0.415 | 0.333 | 0.304 | 0.361 | 0.265 | 0.239 | 0.131 | 0.333 | 0.303 | 0.442 | 0.571 | 0.391 |
| claude-sonnet-4-20250514 | 0.691 | 0.741 | 0.720 | 0.333 | 0.739 | 0.784 | 0.483 | 0.691 | 0.206 | 0.722 | 0.727 | 0.747 | 0.794 | 0.703 |
| deepseek-chat-v3-0324 | 0.596 | 0.593 | 0.652 | 0.333 | 0.565 | 0.649 | 0.404 | 0.568 | 0.191 | 0.630 | 0.667 | 0.693 | 0.750 | 0.562 |
| fanar-c-1-8.7b | 0.415 | 0.309 | 0.499 | 0.500 | 0.565 | 0.392 | 0.258 | 0.355 | 0.152 | 0.500 | 0.364 | 0.496 | 0.621 | 0.391 |
| gemini-2.0-flash | 0.645 | 0.679 | 0.698 | 0.500 | 0.652 | 0.701 | 0.483 | 0.668 | 0.193 | 0.630 | 0.636 | 0.702 | 0.772 | 0.648 |
| gpt-4.1 | 0.681 | 0.580 | 0.712 | 0.333 | 0.652 | 0.711 | 0.404 | 0.653 | 0.197 | 0.722 | 0.561 | 0.757 | 0.794 | 0.656 |
| gpt-5 | 0.795 | 0.778 | 0.768 | **0.667** | 0.739 | **0.866** | **0.709** | **0.792** | 0.215 | 0.759 | **0.864** | 0.822 | 0.810 | **0.766** |
| gpt-o3 | **0.797** | **0.802** | **0.792** | 0.500 | **0.783** | 0.835 | 0.689 | 0.772 | **0.215** | 0.759 | **0.864** | **0.823** | **0.835** | 0.727 |
| llama-3.1-8b-instruct | 0.335 | 0.296 | 0.332 | 0.167 | 0.348 | 0.175 | 0.225 | 0.286 | 0.116 | 0.315 | 0.227 | 0.349 | 0.478 | 0.328 |
| llama-3.3-70b-instruct | 0.487 | 0.556 | 0.585 | 0.333 | 0.435 | 0.577 | 0.325 | 0.506 | 0.177 | 0.463 | 0.576 | 0.598 | 0.695 | 0.539 |
| med42-8b | 0.304 | 0.235 | 0.348 | 0.167 | 0.043 | 0.309 | 0.185 | 0.220 | 0.107 | 0.389 | 0.242 | 0.321 | 0.475 | 0.297 |
| medgemma-4b-it | 0.365 | 0.272 | 0.380 | 0.500 | 0.391 | 0.247 | 0.305 | 0.375 | 0.125 | **0.574** | 0.273 | 0.424 | 0.549 | 0.320 |
| qwen-plus | 0.579 | 0.630 | 0.658 | 0.500 | 0.478 | 0.670 | 0.397 | 0.544 | 0.193 | 0.593 | 0.636 | 0.682 | 0.766 | 0.648 |

Table E2: Overall Model Accuracy per Difficulty Level

| Level | allam-7b-instruct | apollo-7b | c4ai-command-r7b-arabic-02-2025 | claude-sonnet-4-20250514 | deepseek-chat-v3-0324 | fanar-c-1-8.7b | gemini-2.0-flash | gpt-4.1 | gpt-5 | gpt-o3 | llama-3.1-8b-instruct | llama-3.3-70b-instruct | med42-8b | medgemma-4b-it | qwen-plus |
|---|---|---|---|---|---|---|---|---|---|---|---|---|---|---|---|
| Y1 | 0.475 | 0.263 | 0.418 | 0.707 | 0.648 | 0.488 | 0.667 | 0.699 | **0.769** | 0.773 | 0.366 | 0.566 | 0.339 | 0.416 | 0.644 |
| Y2 | 0.430 | 0.215 | 0.350 | 0.664 | 0.578 | 0.433 | 0.618 | 0.635 | 0.723 | **0.730** | 0.324 | 0.510 | 0.322 | 0.357 | 0.581 |
| Y3 | 0.367 | 0.172 | 0.306 | 0.548 | 0.490 | 0.341 | 0.534 | 0.485 | **0.664** | 0.652 | 0.253 | 0.439 | 0.244 | 0.297 | 0.490 |
| Y4 | 0.439 | 0.179 | 0.291 | 0.745 | 0.582 | 0.372 | 0.673 | 0.638 | **0.770** | 0.750 | 0.316 | 0.571 | 0.260 | 0.321 | 0.638 |
| Y5 | 0.316 | 0.184 | 0.253 | 0.698 | 0.562 | 0.347 | 0.656 | 0.667 | **0.799** | 0.781 | 0.271 | 0.507 | 0.226 | 0.354 | 0.549 |

# F    ANSWER CHOICE DISTRIBUTION BALANCE

After constructing the dataset, we observed small but notable imbalances in answer distributions after constructing our dataset.

This could lead to model bias toward any specific answer position (e.g., always selecting "A"). As such, to address this, we analyzed and adjusted the distribution of correct answer choices across all subsets and splits by making minimal targeted adjustments (e.g., reordering options when possible) to bring the correct answer frequencies closer to uniformity. This refinement was done independently for the training and test sets across each answer format group (ABCD, ABCDE, ABCDEF). This adjustment ensures a balanced representation of correct answers and helps reduce the likelihood that models learn position-based heuristics.

Figures F1–F3 visualize the resulting distributions.

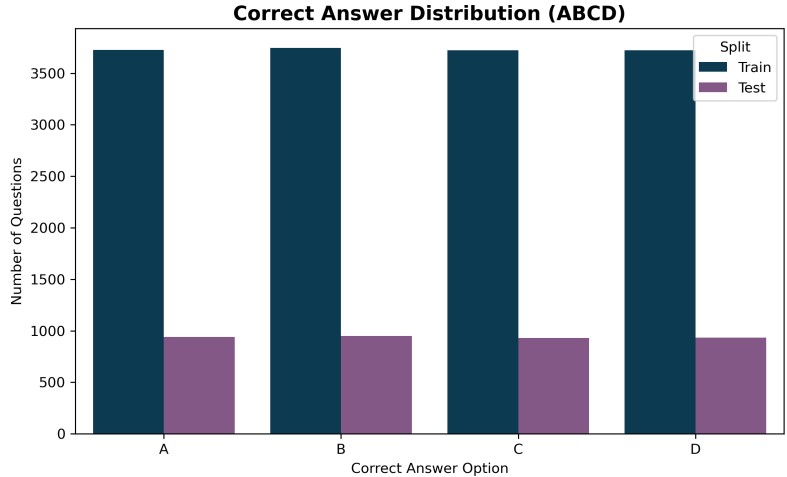

Figure F1: Answer Choice Distribution for ABCD format (Train/Test)

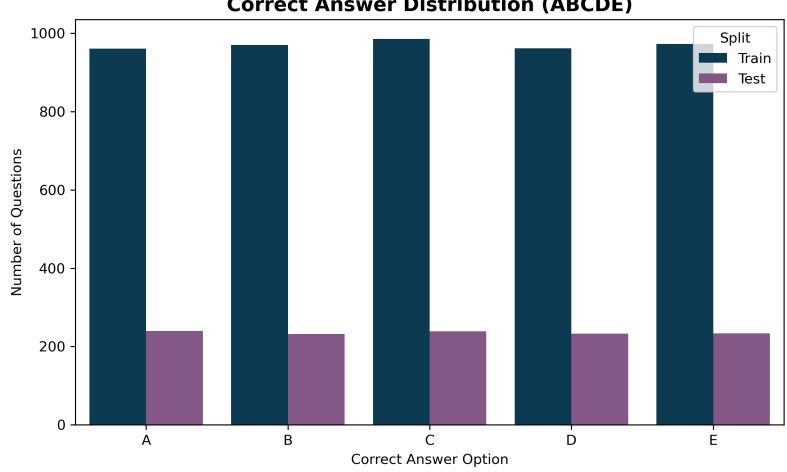

Figure F2: Answer Choice Distribution for ABCDE format (Train/Test)

We further validated this balance using chi-square goodness-of-fit tests. Chi-square goodness-of-fit tests confirmed no significant deviation from uniformity in ABCD and ABCDE splits ($p > 0.97$), indicating that the observed answer distributions do not significantly deviate from a uniform distribution. The ABCDEF format was excluded from

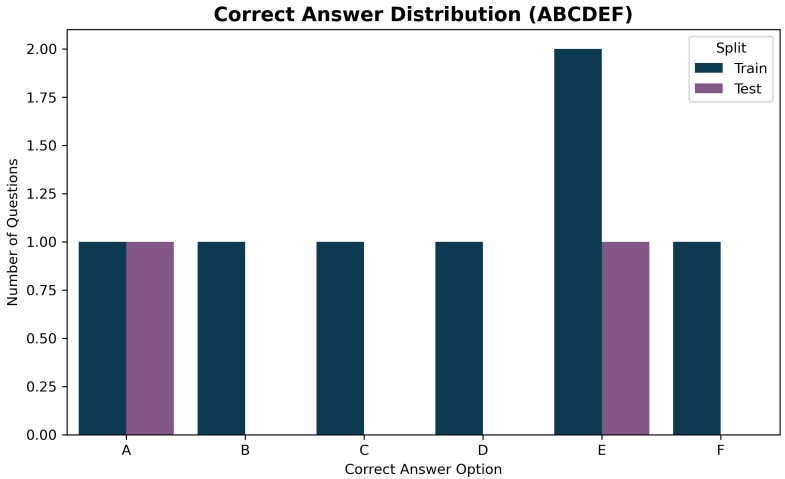

Figure F3: Answer Choice Distribution for ABCDEF format (Train/Test)

this analysis in the test set due to insufficient sample size (n = 2). All tests support the conclusion that the distributions do not significantly differ from uniformity.

The resulting $\chi^2$ values and $p$-values were as follows:

- **ABCD (Train)**: $\chi^2 = 0.099$, $p = 0.992$
- **ABCD (Test)**: $\chi^2 = 0.212$, $p = 0.976$
- **ABCDE (Train)**: $\chi^2 = 0.422$, $p = 0.981$
- **ABCDE (Test)**: $\chi^2 = 0.226$, $p = 0.994$
- **ABCDEF (Train)**: $\chi^2 = 0.714$, $p = 0.982$
- **ABCDEF (Test)**: $\chi^2 = 6.02$, $p = 0.304$

This adjustment ensures a balanced representation of correct answers and helps reduce the likelihood that models learn position-based heuristics.

# G  Contemporary vs Legacy Model Assessment

In this section, we evaluate the evolution of contemporary models against their legacy counterparts by evaluating them on the MedArabiQ Abu Daoud et al. (2025) dataset and comparing their accuracy scores. Additionally, this analysis allows us to compare model scores on the MedArabiQ and MedAraBench benchmarks, providing valuable insights into the value of each benchmark.

The results are shown in Table G1 below. Our experimental results on the MedArabiQ benchmark show that all contemporary models outperform legacy models on the MCQ task. Additionally, our results show that gemini-2.0-flash, gpt-4.1, gpt-5, gpt-o3, and qwen-plus perform better on the MedArabiQ benchmark, while claude-sonnet-4-20250514 and llama-3.3-70b-instruct perform better on MedAraBench. This indicates that the MedAraBench benchmark is more challenging overall, and thus provides larger value for the advancement of medical Arabic NLP. It is important to note that legacy models gemini-1.5-pro and claude-3.5-sonnet were deprecated, which prevented us from evaluating them on MedAraBench to provide a better comparison between benchmarks.

Table G1: Legacy vs contemporary model performance on MedArabiQ and MedAraBench

| MedArabiQ | | | | MedAraBench | |
| --- | --- | --- | --- | --- | --- |
| Legacy Model | Accuracy | Contemporary Model | Accuracy | Contemporary Model | Accuracy |
| claude-sonnet-3.5 | 0.535 | claude-sonnet-4-20250514 | 0.6869 | claude-sonnet-4-20250514 | **0.6937** |
| gemini-1.5-pro | 0.575 | gemini-2.0-flash | **0.7273** | gemini-2.0-flash | 0.6539 |
| | | gpt-4.1 | **0.8081** | gpt-4.1 | 0.6733 |
| gpt-4 | 0.535 | gpt-5 | **0.8586** | gpt-5 | 0.7642 |
| | | gpt-o3 | **0.8384** | gpt-o3 | 0.7652 |
| llama-3.1-8b | 0.262 | llama-3.3-70b-instruct | 0.4949 | llama-3.3-70b-instruct | **0.5466** |
| qwen-2.5-7b | 0.380 | qwen-plus | **0.6566** | qwen-plus | 0.6177 |

