# OpenReview forum: "MedAraBench: Large-scale Arabic Medical Question Answering Dataset and Benchmark"
_ICLR.cc/2026/Conference — ICLR 2026 Poster_

### Official Review · Reviewer_BRmF · 2025-10-30

**Soundness:** 1
**Presentation:** 2
**Contribution:** 2
**Rating:** 2
**Confidence:** 4

**Summary:**

This paper introduces MedAraBench, a large-scale Arabic medical QA benchmark of 24K multiple-choice questions across 19 specialties and five difficulty levels. The dataset was digitized from Arabic medical exams, manually filtered, and evaluated by human experts and through an “LLM-as-a-judge” framework. Eight LLMs (open-source and proprietary) are benchmarked in zero-shot settings. The goal is to provide a specialty-diverse Arabic medical QA benchmark for evaluating LLMs.

**Strengths:**

1. Significant manual data cleaning and digitization effort, which adds credibility and quality to the dataset.

2. Diverse specialty coverage and structured annotation across difficulty levels, ensuring representativeness within the medical domain.

3. Inclusion of a human expert evaluation component, which is commendable and adds qualitative depth to the study.

4. Contributes to Arabic NLP, a domain with limited existing benchmarks and resources.

**Weaknesses:**

1. Unjustified selection of evaluator LLMs (Section 3.2.2)
The paper provides no justification for the selection of the three LLMs used as evaluators in the LLM-as-a-judge setup. There is no discussion of why these particular models were chosen, nor any rationale for excluding medical or Arabic-specialized LLMs, such as BiMediX (arabic+medical) or Fanar(arabic) or medgemma(medical+multilingual) etc .
 A more rigorous approach would have been to compare multiple candidate evaluators and measure their correlation with human expert scores (e.g., Pearson or Spearman coefficients) to identify which LLM aligns best with human judgment.

2. Missing mention of existing Arabic medical benchmarks (Table 1, lines 058–059).
The comparison table omits BiMediX (arXiv:2402.13253), which already provides Arabic translations of MedQA and MedMCQA.

3. Unsupported validation and overstated conclusions (lines 360–362, Table D2)
The conclusion that “ the potential of LLMs to be used for data quality evaluation in the medical domain” (lines 362–363) is overstated and empirically unsupported. The only evidence presented is a superficial similarity in average scores between human and LLM evaluations (Table D2). However, this does not constitute proof of agreement or reliability.
Several methodological issues invalidate the comparison:
- Different evaluation scales: Human experts used a binary high/low rating, while LLMs used a 1–5 Likert scale, making numerical averages non-comparable.
- Different samples: Humans and LLMs did not evaluate the same subset of data
- No agreement metrics: No statistical measure of correlation or agreement between human evaluators and the LLM judges (e.g., Pearson, Spearman, or Cohen’s κ) is reported.
- Moreover, the statement in lines 360–362 that : “While they are not directly comparable due to varying evaluation scales, we note that the results of LLM-as-a-judge and expert quality evaluation are comparable.” is internally contradictory comparability cannot be claimed if the scales and samples differ.
The clause “pending further alignment with medical standards” implicitly acknowledges this weakness, but does not substitute for empirical validation.

4. Unjustified use of the Likert 1–5 scale
The 1–5 scale is applied without defining intermediate values (2–4) in the prompt (Appendix C), and no rationale is provided for using a 5-point scale instead of a binary one matching the human evaluation. This undermines comparability and interpretability.

5. Absence of medical Arabic LLM baselines (Section 3.3)
Although the work benchmarks several proprietary and open models (GPT-5, Gemini, Claude, etc.), no Arabic or medical LLMs are tested, despite the availability of models like BiMediX (Arabic and medical), medgemma (medical and multilingual) etc.

6. Invalid cross-benchmark and cross-model comparisons and flawed analysis of model progress (discussion lines 384–394; Table D1)
In the discussion (lines 384–394), the authors claim to observe “evolution of model performance across generations”.
 This analysis is methodologically invalid, as it compares different models on different benchmarks (MedArabiQ vs. MedAraBench).
 Because neither the models nor the datasets are constant, performance differences cannot be attributed to either factor.
Table D1 seems intended to show that MedAraBench might provide a more informative or challenging evaluation, but the comparison is not correctly designed, and the caption does not clarify what the table represents.
 The two columns correspond to different model generations, making them not directly comparable.
This is a missed opportunity:
 If the authors had evaluated the same models on both MedArabiQ and MedAraBench, they could have shown whether the new benchmark is more challenging and thus more valuable.
 Alternatively, testing different generations of the same model family (e.g., GPT-4 vs GPT-5) on MedAraBench would have allowed a valid analysis of progress over time.
 As it stands, the setup conflates dataset variation with model advancement, so conclusions about “model evolution” are unsupported.
 Both the discussion and Table D1 should be revised: either clarify that the comparison is descriptive or conduct controlled, same-model evaluations.

7. Invalid comparison of models (lines 365–367)
The authors conclude that “proprietary models outperform open-source models,” yet the proprietary models are orders of magnitude larger than open ones. Such comparisons are meaningless without controlling for scale.

8. Dataset imbalance (Figure 2a, Table A2)
Over 56 % (Figure 2a, Table A2) of questions are Year-1 level and only 5 % Year-5, resulting in a dataset dominated by basic-science items. This imbalance likely makes the benchmark less challenging and limits its capacity to assess advanced reasoning.

9. Suspicious perfect accuracies without explanation (Table 3)
In Table 3, several models report perfect accuracies (1.0) for the ABCDEF configuration, while scores on other subsets remain between 0.55–0.77.
This sudden jump to perfect accuracy across models is highly suspicious and atypical for medical QA tasks.
No explanation or investigation is provided. The authors should have clarified whether the ABCDEF subset:
- contains very few items (inflating accuracy),
- includes only Year-1 questions (simpler), or
- whether the addition of letter choices (A–F) helped models guess the correct answer (e.g., positional or formatting cues).
Without such clarification, the results appear unreliable and raise concerns about evaluation validity.

10. Limited novelty and under-utilization of the dataset
While the dataset is valuable for Arabic medical NLP, the contribution is incremental rather than conceptual, there is no new evaluation framework or modeling insight beyond prior work (MedArabiQ). The paper advertises ~24K MCQs, yet only ~4.9K test items are actually used in experiments; the ~20K training split is never explored (no fine-tuning, few-shot, or human/LLM evaluation on train). As a result, the empirical scope is limited to the test set, leaving the benchmark largely under-utilized. The most tangible contribution remains the digitization and manual curation of Arabic medical MCQs.

**Questions:**

Questions:

1. Did you test Arabic or medical-specialized LLMs as potential judges?

2. How is the “Average (Fraction of 5)” metric in Table 3 calculated?

3. How do explain the fact that several models reach perfect (1.0) accuracy in the ABCDEF configuration?

4. What does Table D1 intend to represent, benchmark comparison or model evolution?

5. Could you share the prompts used to evaluate the benchmarked models, including input format, language setup, and answer extraction method?

Remarks:

1. Misplaced or unclear citation (line 099).
The citation to the GPT-4 technical report (Achiam et al., 2023) does not logically connect to the preceding sentence. If the authors meant to refer to GPT-4 being evaluated on translated benchmarks, the sentence should be rephrased for clarity.

2. Missing cross-reference (lines 170–171)
In Section 3.1, methodological details are discussed without referencing the appropriate subsection (Section 4.1), reducing readability.

3. Lack of explanation for evaluation platform (line 215)
The authors mention that expert evaluations were conducted using Qualtrics, yet they do not explain what it is nor provide a footnote or citation.

4. Incomplete sentence (line 331)

5.  Invalid link (line 452):
The repository link (https://anony-mous.4open.science/r/medarabench-3BE4/) is inaccessible.

---

> ### Author Response · Authors · 2025-11-23
>
> [1/n]
> Thank you for your thoughtful and constructive feedback! Below, we provide a detailed point-by-point response to each of your questions and comments.
>
> > Unjustified selection of evaluator LLMs (Section 3.2.2) The paper provides no justification for the selection of the three LLMs used as evaluators in the LLM-as-a-judge setup. There is no discussion of why these particular models were chosen, nor any rationale for excluding medical or Arabic-specialized LLMs, such as BiMediX (arabic+medical) or Fanar(arabic) or medgemma(medical+multilingual) etc . A more rigorous approach would have been to compare multiple candidate evaluators and measure their correlation with human expert scores (e.g., Pearson or Spearman coefficients) to identify which LLM aligns best with human judgment.
>
> > Different evaluation scales: Human experts used a binary high/low rating, while LLMs used a 1–5 Likert scale, making numerical averages non-comparable.
>
> Thank you for your extensive feedback on this matter. Please see below:
>
> Originally, our choice of using GPT-4, Gemini-1.5-Pro, and Claude-3.5-Sonnet as our evaluator LLMs was motivated by their superior performance in the existing MedArabiQ benchmark [1]. We agree that our manuscript did not clearly state this.
>
> However, based on your feedback, we have entirely revised our LLM-as-a-judge set up during the rebuttal period and instead we now (i) use the four best-performing models on MedAraBench as multiple candidate evaluators (GPT-o3, Gemini-2.0-Flash, Claude-4-Sonnet, GPT-5), (noting that we also benchmarked new Arabic/medical models but will discuss those later) (ii) modified the evaluation setup to a binary scale for improved comparability with expert ratings, and (iii) assessed correlation with human expert scores. Our revised approach ensures that model-based evaluation reflects state-of-the-art performance for Arabic and medical content in this context. Detailed results are in the next comments.
>
> Below are the updated results for multiple candidate evaluators (best performing models) using the binary scale:
>
> Table 1 - LLM-as-a-judge Experiments Using the Top Performing Models on the MedAraBench Dataset
> | Model             | Medical Accuracy | Clinical Relevance | Question Quality | Question Difficulty |
> |-------------------|------------------|---------------------|------------------|----------------------|
> | GPT-o3            | 0.673 [0.469]    | 0.827 [0.378]       | 0.588 [0.492]    | 0.841 [0.366]        |
> | Gemini-2.0-Flash  | 0.717 [0.450]    | 0.565 [0.496]       | 0.815 [0.388]    | 0.774 [0.418]        |
> | Claude-4-Sonnet   | 0.711 [0.453]    | 0.749 [0.434]       | 0.576 [0.494]    | 0.764 [0.425]        |
> | GPT-5             | 0.533 [0.499]    | 0.610 [0.488]       | 0.597 [0.490]    | 0.476 [0.499]        |
>
> In order to properly evaluate LLMs as evaluators, we have analyzed Pearson Correlation scores of expert and LLM evaluations on the same 378-question subset of the dataset, and found that GPT-o3 was the best-performing model in terms of alignment with expert evaluations. The results of this experiment are shown in Table 2 below:
>
> Table 2 - Pearson Correlation Scores of LLM Evaluators with Expert Evaluators
> | Assessment Criterion | GPT-o3 (A) | Claude-4-Sonnet (A) | Gemini-2.0-Flash (A) | GPT-5 (A) | GPT-o3 (B) | Claude-4-Sonnet (B) | Gemini-2.0-Flash (B) | GPT-5 (B) |
> |----------------------|------------|-----------------------|------------------------|-----------|------------|-----------------------|------------------------|-----------|
> | Medical Accuracy     | 0.577      | 0.065                 | 0.023                  | 0.043     | 0.505      | 0.053                 | 0.053                  | 0.068     |
> | Clinical Relevance   | 0.252      | 0.165                 | 0.176                  | 0.071     | 0.377      | 0.131                 | 0.131                  | 0.104     |
> | Question Quality     | 0.407      | 0.007                 | 0.023                  | 0.044     | 0.336      | 0.062                 | 0.062                  | 0.033     |
> | Question Difficulty  | -0.114     | -0.070                | 0.019                  | -0.116    | -0.018     | 0.039                 | 0.039                  | -0.049    |

---

> ### Author Response · Authors · 2025-11-23
>
> [2/n]
> > Different samples: Humans and LLMs did not evaluate the same subset of data
>
> We acknowledge that the LLMs were evaluated a larger subset of the test set compared to the expert evaluators. This design choice reflects practical resource constraints, whereas a more rigorous human evaluation would require substantially more time and effort. To ensure fairness and minimize bias, the subset assessed by human experts was selected randomly from the full test set evaluated by Additionally, we conducted statistical analyses on this expert-reviewed subset to assess the alignment between LLM and human ratings. We have now updated the results as follows:
>
> - Our newly selected LLMs evaluate the entire test set, the results of this analysis are shown in Table 3 of the updated manuscript.
> - LLMs and human annotators evaluate the same subset, and we measure correlation on this subset. The results of this analysis are shown in Table C1 in Appendix C of the updated manuscript.
> - We selected gpt-o3 as it has the highest correlation on the human-annotated subset to be evaluated on the full training set, as requested by the reviewer. This will be updated in Appendix C as soon as the analysis is complete.
>
> > Unsupported validation and overstated conclusions (lines 360–362, Table D2) The conclusion that “ the potential of LLMs to be used for data quality evaluation in the medical domain” (lines 362–363) is overstated and empirically unsupported. The only evidence presented is a superficial similarity in average scores between human and LLM evaluations (Table D2). However, this does not constitute proof of agreement or reliability.
>
> We agree with the reviewer, and modified the old statement given that our experiments were rerun on a binary evaluation scale, and added the following text in the Discussion section of the updated manuscript:
>
> *“We further evaluate the evolution of models across generations by comparing contemporary and legacy model performance on the MedArabiQ benchmark. Our results, presented in Table F1 in Appendix F below, show that all contemporary models outperform legacy models on the MCQ task. Additionally, this analysis allows us to compare the merit of both the MedArabiQ and MedAraBench benchmarks, with gemini-2.0-flash, gpt-4.1, gpt-5, gpt-o3, and qwen-plus performing better on the MedArabiQ benchmark, while claude-sonnet-4-20250514 and llama-3.3-70b-instruct perform better on MedAraBench. This indicates that the MedAraBench benchmark is more challenging overall, highlighting its value in advancing medical Arabic NLP.”*
>
> > How is the “Average (Fraction of 5)” metric in Table 3 calculated?
>
> We have updated Table 3 with the binary LLM-as-a-judge experiment results, thus omitting the “Average (Fraction of 5)” metric from the paper.
>
> > Did you test Arabic or medical-specialized LLMs as potential judges?
>
> We opted to choose the best-performing models as our evaluation judges, given their strong and well-documented performance in multilingual and medical reasoning tasks across a wide range of benchmarks. Our goal in this setup was to approximate a high-quality, model-based proxy for expert judgment rather than to evaluate the judging capability of Arabic- or medical-specialized LLMs themselves. Exploring specialized Arabic or medical LLMs as potential judges is an interesting direction, but it is beyond the scope of the present work, which focuses on using strong frontier models to assess the quality of MedAraBench.

---

> ### Author Response · Authors · 2025-11-23
>
> [3/n]
> > Absence of medical Arabic LLM baselines (Section 3.3) Although the work benchmarks several proprietary and open models (GPT-5, Gemini, Claude, etc.), no Arabic or medical LLMs are tested, despite the availability of models like BiMediX (Arabic and medical), medgemma (medical and multilingual) etc.
>
> Thank you for your feedback on this matter. We agree with your comment, and have performed additional benchmarking experiments using the following models and their respective overall accuracy scores:
> - General Purpose Models
>    - LLaMa-3.1-8B (17.0%)
> - Arabic-centric Models
>    - Fanar-C-1-8.7B (49.77%)
>    - Allam-7B-instruct (44.6%)
>    - Cohere c4ai-command-r7B-arabic-02-2025 (38.07%)
> - Medical Multilingual Models
>    - Medgemma-4B-it (39.02%)
>    - Apollo-7B (23.78%)
>    - Med42-8B (31.82%)
>
> The additional results provide valuable information on how Arabic-centric and medical multilingual models compare to general-purpose LLMs on MedAraBench, and highlight that even specialized models still underperform proprietary systems on this task. We have integrated these baselines into section 4.4 and updated Table 4 (and corresponding appendix tables) to report their accuracies alongside the previously evaluated models. We have also updated our discussion to analyze these results in light of model size, architecture, and training focus, noting that smaller or domain-specialized models do not consistently close the gap with larger proprietary models, especially on more challenging subsets.
>
> *Note:* We benchmarked Meditron, however, the results were not meaningful. Our assumption is that the model does not process Arabic. In the original Meditron paper, the authors adapt LLama-2 using primarily English data and then fine-tune and test with each specific MCQ training/validation dataset. Hence, it is out of scope for our work. As for BiMediX, we are running the model and will share the new results in the next few days.

---

> > ### Author Response · Authors · 2025-11-23
> >
> > [4/n]
> > > Invalid cross-benchmark and cross-model comparisons and flawed analysis of model progress (discussion lines 384–394; Table D1) In the discussion (lines 384–394), the authors claim to observe “evolution of model performance across generations”. This analysis is methodologically invalid, as it compares different models on different benchmarks (MedArabiQ vs. MedAraBench). Because neither the models nor the datasets are constant, performance differences cannot be attributed to either factor. Table D1 seems intended to show that MedAraBench might provide a more informative or challenging evaluation, but the comparison is not correctly designed, and the caption does not clarify what the table represents. The two columns correspond to different model generations, making them not directly comparable. This is a missed opportunity: If the authors had evaluated the same models on both MedArabiQ and MedAraBench, they could have shown whether the new benchmark is more challenging and thus more valuable. Alternatively, testing different generations of the same model family (e.g., GPT-4 vs GPT-5) on MedAraBench would have allowed a valid analysis of progress over time. As it stands, the setup conflates dataset variation with model advancement, so conclusions about “model evolution” are unsupported. Both the discussion and Table D1 should be revised: either clarify that the comparison is descriptive or conduct controlled, same-model evaluations.
> >
> > Thank you for your thorough feedback on this matter. We agree with your comments, and have updated our experiments by running contemporary models on MedArabiQ, allowing for a fair comparison between contemporary and legacy models, as well as an indication on which benchmark among MedArabiQ and MedAraBench is more challenging. The results are as shown in the Table below:
> >
> > | Legacy Model         | MedArabiQ Accuracy | Contemporary Model        | MedArabiQ Accuracy | MedAraBench Accuracy |
> > |----------------------|--------------------|----------------------------|-----------------------|----------|
> > | claude-sonnet-3.5    | 53.5               | claude-sonnet-4-20250514   | 0.687          | 0.694 |
> > | gemini-1.5-pro       | 57.5               | gemini-2.0-flash           | 0.727          | 0.654 |
> > | gpt-4                | 53.5               | gpt-4.1                    | 0.808          | 0.673 |
> > |  -               | -                 | gpt-5                      | 0.859         | 0.764|
> > | -               | -                  | gpt-o3                     | 0.838          | 0.765 |
> > | llama-3.1-8b         | 26.2               | llama-3.3-70b-instruct     | 0.495        | 0.547 |
> > | qwen-2.5-7b          | 38                 | qwen-plus                  | 0.657         | 0.618  |
> >
> > Our experimental results on the MedArabiQ benchmark show that all contemporary models outperform legacy models on the MCQ task. Additionally, our results show that gemini-2.0-flash, gpt-4.1, gpt-5, gpt-o3, and qwen-plus perform better on the MedArabiQ benchmark, while claude-sonnet-4-20250514 and llama-3.3-70b-instruct perform better on MedAraBench. This indicates that the MedAraBench benchmark is more challenging overall, and thus provides larger value for the advancement of medical Arabic NLP.
> > It is important to note that legacy models gemini-1.5-pro and claude-3.5-sonnet were deprecated, which prevented us from evaluating them on MedAraBench to provide a better comparison between benchmarks.
> >
> > > What does Table D1 intend to represent, benchmark comparison or model evolution?
> >
> > Originally, Table D1 was meant to represent model evolution. However, we have updated Table D1 in the manuscript to mirror the Table above, which allows us to study both model evolution and a benchmark comparison between MedArabiQ and MedAraBench.
> >
> > > Invalid comparison of models (lines 365–367) The authors conclude that “proprietary models outperform open-source models,” yet the proprietary models are orders of magnitude larger than open ones. Such comparisons are meaningless without controlling for scale.
> >
> > Initially, our model grouping as “proprietary” and “open-source” was due to the fact that open-source models can offer more transparency, and thus allow for adaptation to a variety of tasks at a much larger freedom compared to closed-source models. The distinction was used to organize our experiments across a diverse landscape of models that differ widely in architecture, scale, and training data. We acknowledge this heterogeneity and have thus updated Table 4 in the Results section to categorize models according to three categories (General Purpose, Arabic Centric, Medical). Additionally, we included training dataset size in Table 4 to allow for a clear comparison of baseline results on the MedAraBench dataset.

---

> > > ### Author Response · Authors · 2025-11-23
> > >
> > > [5/n]
> > > > Dataset imbalance (Figure 2a, Table A2) Over 56 % (Figure 2a, Table A2) of questions are Year-1 level and only 5 % Year-5, resulting in a dataset dominated by basic-science items. This imbalance likely makes the benchmark less challenging and limits its capacity to assess advanced reasoning.
> > >
> > > We acknowledge the reviewer’s concern regarding the skew toward Year-1 questions relative to other difficulty levels. This distribution is due to the availability of source materials, which contained disproportionately more early-year content. While this evaluation limits the benchmark’s capacity to assess advanced clinical reasoning, it establishes a baseline performance for Arabic medical understanding across the present difficulty levels and specialties, revealing that even basic medical reasoning remains challenging for current LLMs.
> > >
> > > In light of this observation, we performed further analysis to study model accuracy variation across levels and found that all models exhibited an accuracy drop from Y1 to Y3 before recovering in Y4-Y5. This shows that models exhibit a fairly consistent performance in early and late years, demonstrating the merit of assessing both basic-science and advanced items. Additionally, this consistent pattern indicates that Y3 questions, which likely require more complex clinical reasoning, present the greatest challenge across all model architectures.
> > >
> > > The significant performance variation between difficulty levels highlights critical limitations in current LLMs for medical applications. The average 15-20% performance drop from Y1 to Y3 suggests that models struggle with the integrative clinical reasoning required at intermediate levels. This difficulty scaling effect was most pronounced for smaller models, indicating that scale and sophisticated training are crucial for handling complex medical reasoning tasks. These findings suggest LLMs may be better suited for foundational knowledge and specialized domains than for complex clinical decision-making.
> > >
> > >  We elaborate further on this matter in Section 6 (lines 461 - 468) and Appendix D, and acknowledge that future work should prioritize the collection of advanced Arabic medical content.
> > >
> > > > Suspicious perfect accuracies without explanation (Table 3) In Table 3, several models report perfect accuracies (1.0) for the ABCDEF configuration, while scores on other subsets remain between 0.55–0.77. This sudden jump to perfect accuracy across models is highly suspicious and atypical for medical QA tasks. No explanation or investigation is provided. The authors should have clarified whether the ABCDEF subset: contains very few items (inflating accuracy), includes only Year-1 questions (simpler), or whether the addition of letter choices (A–F) helped models guess the correct answer (e.g., positional or formatting cues). Without such clarification, the results appear unreliable and raise concerns about evaluation validity.
> > >
> > > > How do explain the fact that several models reach perfect (1.0) accuracy in the ABCDEF configuration?
> > >
> > > There is a total of 25,694 questions in the MedAraBench dataset. The data subsets are split as follows:
> > >
> > > - ABCD: 19,444 questions (75.67% of the dataset)
> > > - ABCDE: 6,241 questions (24.29% of the dataset)
> > > - ABCDEF: 9 questions (0.04% of the dataset)
> > >
> > > The small size of the ABCDEF subset led to the inflation of accuracy scores in the benchmarking experiments for all models. As such, we appreciate your feedback and have omitted the ABCDEF subset as its own category in our analysis and graphs.

---

> > > > ### Author Response · Authors · 2025-11-23
> > > >
> > > > [6/n]
> > > > > Limited novelty and under-utilization of the dataset While the dataset is valuable for Arabic medical NLP, the contribution is incremental rather than conceptual, there is no new evaluation framework or modeling insight beyond prior work (MedArabiQ). The paper advertises ~24K MCQs, yet only ~4.9K test items are actually used in experiments; the ~20K training split is never explored (no fine-tuning, few-shot, or human/LLM evaluation on train). As a result, the empirical scope is limited to the test set, leaving the benchmark largely under-utilized. The most tangible contribution remains the digitization and manual curation of Arabic medical MCQs.
> > > >
> > > > Thank you for your feedback on this matter. We agree that we could further explore fine-tuning and few-shot learning, which is the main purpose of releasing this training set. As such, we have performed few-shot and QLoRA experiments on LLaMa-3.1-8B to fully explore the utility of our dataset in Arabic medical NLP. Our few-shot experiments were performed by providing 3 sample questions rated high across all 4 metrics (question quality, clinical relevance, question difficulty, and medical accuracy) by both expert evaluators to ensure high-quality finetuning. The chosen questions were omitted from the test set and are shown in Appendix B of the updated manuscript
> > > >
> > > > As for our LoRA experiments, we used the training split to fine-tune the Llama-3.1-8b-instruct model using parameter-efficient QLoRA in 4-bit precision. We prepared the training data with prompt-response pairs in Arabic, formatted for multiple-choice questions. The model was loaded in quantized mode and LoRA adapters were applied to key attention modules (q_proj, k_proj, v_proj, o_proj, v_proj), with a training run for up to 800 steps using batched gradient accumulation and standard regularization. This allowed us to efficiently adapt our models to Arabic medical MCQs and directly assess the impact of MedAraBench data on open-source model performance. The results of both few-shot and LoRA experiments are shown in the Table below and are reported in Section 4.5 of the manuscript.
> > > >
> > > > | Model                  | Baseline Accuracy | Fewshot Accuracy | LoRA Accuracy |
> > > > |------------------------|------------------|------------------|---------------|
> > > > | llama-3.1-8b-instruct  | 0.170            | 0.191            | 0.320         |
> > > >
> > > > The experiments revealed substantial improvements over the baseline zero-shot performance for Llama-3.1-8B-instruct, though with notable differences in effectiveness between the two approaches. Few-shot learning provided a modest gain of 12.4% (from 0.170 to 0.191), suggesting that in-context examples helped the model better understand the medical question format and reasoning patterns. Additionally, QLoRA fine-tuning boosted accuracy by 88.2% to 0.320, nearly doubling the model's performance. This is further discussed in the results section of our updated manuscript.
> > > >
> > > > References:
> > > > 1. Z. Huang, W. Zhu, G. Cheng, L. Li, and F. Yuan, “MindMerger: Efficient Boosting LLM Reasoning in non-English Languages,” arXiv.org, 2024. https://arxiv.org/abs/2405.17386 .
> > > >
> > > > > Could you share the prompts used to evaluate the benchmarked models, including input format, language setup, and answer extraction method?
> > > >
> > > > The entire benchmarking strategy is shown below:
> > > > - Prompt:
> > > >
> > > > *"You are an expert medical virtual assistant. Please provide the correct answer letter (A, B, C, or D) for the following Arabic medical multiple-choice question.*
> > > >
> > > > *Question: {question_text_in_arabic}*
> > > >
> > > > *Options:*
> > > >
> > > > *A: {option_a_in_arabic}*
> > > >
> > > > *B: {option_b_in_arabic}*
> > > >
> > > > *C: {option_c_in_arabic}*
> > > >
> > > > *D: {option_d_in_arabic}*
> > > >
> > > > *Answer:"*
> > > >
> > > > - Answer Extraction: We implemented MCQ benchmarking using answer extraction via post-processing. Models were prompted to output the answer choice letter (A-D) directly, and we parsed their text responses using pattern matching to extract the answer.
> > > > - Language Setup: We did not explicitly set language parameters in API calls since models automatically detect Arabic text from the input.
> > > >
> > > > We have updated Section 3.3 to include the benchmarking strategy in the manuscript.

---

> > > > > ### Author Response · Authors · 2025-11-23
> > > > >
> > > > > [7/7]
> > > > > > Misplaced or unclear citation (line 099). The citation to the GPT-4 technical report (Achiam et al., 2023) does not logically connect to the preceding sentence. If the authors meant to refer to GPT-4 being evaluated on translated benchmarks, the sentence should be rephrased for clarity.
> > > > >
> > > > > > Missing cross-reference (lines 170–171) In Section 3.1, methodological details are discussed without referencing the appropriate subsection (Section 4.1), reducing readability.
> > > > >
> > > > > > Lack of explanation for evaluation platform (line 215) The authors mention that expert evaluations were conducted using Qualtrics, yet they do not explain what it is nor provide a footnote or citation.
> > > > >
> > > > > > Incomplete sentence (line 331)
> > > > >
> > > > > > Invalid link (line 452): The repository link (https://anony-mous.4open.science/r/medarabench-3BE4/) is inaccessible.
> > > > >
> > > > > Thank you for pointing out these errors.
> > > > > - We fixed the citation on line 099.
> > > > > - We added the missing cross-references on lines 170-171, referencing Appendix A for better readability on the detailed breakdown of our dataset.
> > > > > - We provided an explanation on Qualtrics as an evaluation platform and added a citation on line 215.
> > > > > - We fixed the incomplete sentence on line 331.
> > > > > - We provide a valid repository link on line 452 (https://anonymous.4open.science/r/medarabench-3BE4/README.md).

---

> ### Author Response · Authors · 2025-11-27
>
> [UPDATE]
>
> Dear Reviewer,
>
> Following on the above, we would like to present further results based on your requests for the LLM-as-a-Judge evaluation on the training set and benchmarking BiMedix-Bi as the experiments just concluded:
>
> - We completed LLM-as-a-Judge evaluations over the entire training set using GPT-o3. We selected GPT-o3 because it achieved the best Pearson Correlation scores with expert ratings across all four evaluation metrics on the expert-annotated subset (Table C1, Appendix C). Additionally, we ran t-tests to compare GPT-o3 scores on the training and test sets and found no significant difference on all 4 evaluation metrics, indicating consistent quality across splits. The results of this experiment are shown in Appendix D of the updated manuscript.
> - We extended our benchmarking to include BiMedix-Bi, providing a more comprehensive comparison between general-purpose, Arabic-centric, and medical LLMs on MedAraBench. The model scored a baseline accuracy of 0.390 on our test set. The results of this experiment are shown in Table 4 and Figure 1 of the updated manuscript.
>
> We kindly ask you to take the above updates into consideration when evaluating our submission.
> Thank you for your time, and we look forward to hearing from you in case of any further questions.

---

### Official Review · Reviewer_m6wR · 2025-10-30

**Soundness:** 2
**Presentation:** 2
**Contribution:** 2
**Rating:** 4
**Confidence:** 4

**Summary:**

This paper introduces MedAraBench, a benchmark for evaluating LLMs on Arabic medical MCQ task. The dataset contains about 24k questions across 19 medical domains and 5 difficulty levels, manually digitized from Arabic medical school materials. The authors conduct both clinical expert evaluation and LLM-as-a-judge assessments to measure data quality, finding moderate agreement and generally acceptable accuracy. They then benchmark open-source and proprietary models like GPT-5, Gemini 2.0, and Claude 4, showing that proprietary models outperform open-source ones but still fall short of expert level accuracy. The paper provides a structured resource for testing Arabic medical tasks but is mainly limited to multiple-choice formats and zero-shot evaluations.

**Strengths:**

- The manual digitization and expert validation of data from non-digital academic sources shows significant effort and ensure the dataset’s authenticity and reliability.
- The dataset spans 19 medical specialties at various difficulty levels, offering a structured framework that supports fine grained evaluation of LLM performance across various domains of medical knowledge for the Arabic language.

**Weaknesses:**

- For validating data quality using LLM-as-a-judge, the authors employ GPT-4, Gemini 1.5 Pro, and Claude 3.5 Sonnet. However, there is no justification provided for selecting these specific models, Are they known to outperform others in Arabic understanding? Moreover, the prompt instructs the models to act as medical education expert but does not account for the Arabic language aspect of the task. The capability of these models in Arabic medical understanding needs further evaluation.
- The literature review and experimental comparisons are not comprehensive. Open-source models like BiMediX [1] have benchmarked Arabic medical tasks and have released translated and verified datasets. Additionally, compare with more medical open-source models like Apollo [2], Med42 [3], Meditron [4]. Proprietary models like Gemini 2.5 Pro [5] and Flash are also missing from the evaluations.
- If LLM as a judge is an automated framework to evaluate the quality of the proposed dataset. Extending this validation to the training set could help further filter and enhance the quality of the data. Currently, the validation is limited only to the test set.

[1] *Pieri, Sara, et al. "Bimedix: Bilingual medical mixture of experts llm." arXiv preprint arXiv:2402.13253 (2024).*

[2] *Wang, Xidong, et al. "Apollo: A lightweight multilingual medical LLM towards democratizing medical AI to 6B people." arXiv preprint arXiv:2403.03640 (2024).*

[3] *Christophe, Clément, et al. "Med42-v2: A suite of clinical llms." arXiv preprint arXiv:2408.06142 (2024).*

[4] *Chen, Zeming, et al. "Meditron-70b: Scaling medical pretraining for large language models." arXiv preprint arXiv:2311.16079 (2023).*

[5] *Comanici, Gheorghe, et al. "Gemini 2.5: Pushing the frontier with advanced reasoning, multimodality, long context, and next generation agentic capabilities." arXiv preprint arXiv:2507.06261 (2025)*

**Questions:**

Please address the above weaknesses.
- How were the medical specialties determined for the data samples? Was this an automated process or done with the help of clinical experts?
- Line 197: “No Cueing: options do not provide clues to other answers.” Is it meant to be “Clueing”?

---

> ### Author Response · Authors · 2025-11-23
>
> [1/n]
> Thank you for your thoughtful and constructive feedback! Below, we provide a detailed point-by-point response to each of your questions and comments.
>
> > For validating data quality using LLM-as-a-judge, the authors employ GPT-4, Gemini 1.5 Pro, and Claude 3.5 Sonnet. However, there is no justification provided for selecting these specific models, Are they known to outperform others in Arabic understanding? Moreover, the prompt instructs the models to act as medical education expert but does not account for the Arabic language aspect of the task. The capability of these models in Arabic medical understanding needs further evaluation.
>
> Thank you for your feedback on this matter. Originally, our choice of using GPT-4, Gemini-1.5-Pro, and Claude-3.5-Sonnet as our evaluator LLMs was motivated by their superior performance across the MedArabiQ benchmark [1]. We agree that our manuscript did not clearly state this.
>
> However, based on feedback from multiple reviewers, we have entirely revised our LLM-as-a-judge set up during the rebuttal period and instead we now (i) use the four best-performing models on MedAraBench (GPT-o3, Gemini-2.0-Flash, Claude-4-Sonnet, GPT-5) and (ii) modified the evaluation setup to a binary scale for improved comparability with expert ratings. Our revised approach ensures that model-based evaluation reflects state-of-the-art performance for Arabic and medical content in this context. The results of our updated LLM-as-a-judge experiments are as follows:
>
> Table 1 - LLM-as-a-judge Experiments Using the Top Performing Models on the MedAraBench Dataset
>
> | Model             | Medical Accuracy | Clinical Relevance | Question Quality | Question Difficulty |
> |-------------------|------------------|---------------------|------------------|----------------------|
> | GPT-o3            | 0.673 [0.469]    | 0.827 [0.378]       | 0.588 [0.492]    | 0.841 [0.366]        |
> | Gemini-2.0-Flash  | 0.717 [0.450]    | 0.565 [0.496]       | 0.815 [0.388]    | 0.774 [0.418]        |
> | Claude-4-Sonnet   | 0.711 [0.453]    | 0.749 [0.434]       | 0.576 [0.494]    | 0.764 [0.425]        |
> | GPT-5             | 0.533 [0.499]    | 0.610 [0.488]       | 0.597 [0.490]    | 0.476 [0.499]        |
>
> We have updated Section 3.2.2 in the manuscript to better reflect our reasoning: “Motivated by previous literature [MedArabiQ], we prompted four of our best-performing SOTA LLMs (gpt-03, gemini-2.0-flash, and claude-4-sonnet) to act as medical education experts and to evaluate the MCQs along the same metrics used in our expert quality evaluation: Medical Accuracy, Clinical Relevance, Question Difficulty, and Question Quality, on a binary (0 or 1) scale for the entire test set. Additionally, we calculated Pearson Correlation coefficients for each model and the expert reviewers on the 378-question dataset evaluated by our medical experts. This approach gives us the advantage of providing a more nuanced evaluation across a broader set of our data, while also providing insights into the efficacy of LLMs in evaluating the quality of Arabic medical data relative to expert annotators.”
>
> > If LLM as a judge is an automated framework to evaluate the quality of the proposed dataset. Extending this validation to the training set could help further filter and enhance the quality of the data. Currently, the validation is limited only to the test set.
>
> We agree with the suggestion by the reviewer, and have opted to evaluate the entire dataset using GPT-o3 as our evaluator model only, as it is quite expensive to run all models. We selected GPT-o3 due to the fact that it achieved the best Pearson Correlation scores when evaluated with responses from expert evaluators on a small subset of the data relative to the 3 other LLM evaluators, which is shown in Table C1 in Appendix C of the updated manuscript. Given the large size of our training dataset, the evaluations are still ongoing and will be provided during the rebuttal period as soon as they are available.

---

> ### Author Response · Authors · 2025-11-23
>
> [2/2]
> > The literature review and experimental comparisons are not comprehensive. Open-source models like BiMediX [1] have benchmarked Arabic medical tasks and have released translated and verified datasets. Additionally, compare with more medical open-source models like Apollo [2], Med42 [3], Meditron [4]. Proprietary models like Gemini 2.5 Pro [5] and Flash are also missing from the evaluations.
>
> Thank you for your feedback on this matter. We agree with your comment and have performed additional benchmarking experiments using the following models and their respective overall accuracy scores:
> - General Purpose Models
>    - LLaMa-3.1-8B (17.0%)
> - Arabic-centric Models
>    - Fanar-C-1-8.7B (49.77%)
>    - Allam-7B-instruct (44.6%)
>    - Cohere c4ai-command-r7B-arabic-02-2025 (38.07%)
> - Medical Multilingual Models
>    - Medgemma-4B-it (39.02%)
>    - Apollo-7B (23.78%)
>    - Med42-8B (31.82%)
>
> The additional results provide valuable information on how Arabic-centric and medical multilingual models compare to general-purpose LLMs on MedAraBench, and highlight that even specialized models still underperform proprietary systems on this task. We have integrated these baselines into section 4.4 and updated Table 4 (and corresponding appendix tables) to report their accuracies alongside the previously evaluated models. We have also updated our discussion to analyze these results in light of model size, architecture, and training focus, noting that smaller or domain-specialized models do not consistently close the gap with larger proprietary models, especially on more challenging subsets.
>
> *Note:* We benchmarked Meditron, however, the results were not meaningful. Our assumption is that the model does not process Arabic. In the original Meditron paper, the authors adapt LLama-2 using primarily English data and then fine-tune and test with each specific MCQ training/validation datasets. Hence, it is out of scope for our work. As for BiMediX, we are running the model and will share the new results in the next few days.
>
> > How were the medical specialties determined for the data samples? Was this an automated process or done with the help of clinical experts?
>
> The scanned documents collected from medical school repositories were originally categorized by specialty, which allowed us to directly inherit the medical specialty classification for each question. This was reflected on line 185 in Section 3.1 in the updated document for further clarity.
>
> > No Cueing: options do not provide clues to other answers.” Is it meant to be “Clueing”?
>
> Thank you for catching this error. We fixed the typo and changed it to “Clueing”.
> We kindly ask you to evaluate our submission in light of the above feedback and improvements made to our manuscript.
>
> Thank you for your time and effort.

---

> ### Author Response · Authors · 2025-11-27
>
> [UPDATE]
>
> Dear Reviewer,
>
> Following on the above, we would like to present further results based on your requests for the LLM-as-a-Judge evaluation on the training set and benchmarking BiMedix-Bi as the experiments just concluded:
>
> - We completed LLM-as-a-Judge evaluations over the entire training set using GPT-o3. We selected GPT-o3 because it achieved the best Pearson Correlation scores with expert ratings across all four evaluation metrics on the expert-annotated subset (Table C1, Appendix C). Additionally, we ran t-tests to compare GPT-o3 scores on the training and test sets and found no significant difference on all 4 evaluation metrics, indicating consistent quality across splits. The results of this experiment are shown in Appendix D of the updated manuscript.
> - We extended our benchmarking to include BiMedix-Bi, providing a more comprehensive comparison between general-purpose, Arabic-centric, and medical LLMs on MedAraBench. The model scored a baseline accuracy of 0.390 on our test set. The results of this experiment are shown in Table 4 and Figure 1 of the updated manuscript.
>
> We kindly ask you to take the above updates into consideration when evaluating our submission.
> Thank you for your time, and we look forward to hearing from you in case of any further questions.

---

### Official Review · Reviewer_ratc · 2025-11-01

**Soundness:** 3
**Presentation:** 3
**Contribution:** 3
**Rating:** 8
**Confidence:** 3

**Summary:**

The paper presents a benchmark for medical knowledge in Arabic. The dataset comprises multi-choice questions in 19 medical specialities, extracted and refined manually from exams and lecture notes of medical schools in the Arabic-speaking world, then quality-checked by experts and LLM-as-a-judge scheme. The paper presents benchmarking results for a number of SOTA LLMs.

**Strengths:**

The paper presents a useful resource for Arabic medical understanding, and benchmarking results for a number of SOTA models.

**Weaknesses:**

Some missing details about the construction of the dataset and the implementation of the benchmarking are mentioned in Questions below

**Questions:**

- How is it that Arabic is underrepresented in the medical domain because of its rich morphology and dialectal variation? And why the uneven linguistic landscape calls for medical LLMs? The motivation of this work in Sec.1 should be revised.

- The reference to Appendix A is missing in Sec. 3.1

- MedArabiQ is missing from Table 1

- How were lecture notes converted to meaningful MCQs? This is an important detail that deserves to be discussed.

- The LLMs chosen for benchmarking do not include any Arabic-focused model. Why not? The differences in the benchmarking results could be due to the models' inherent limitation in Arabic itself rather than in medical knowledge, where bigger, proprietary models have an edge, but this angle is not explored.

- Line 331 (Sec 4.3) is truncated.

- How big were the subsets of MCQ? Guessing from Fig. E3. subset ABCDEF has only 2 questions in the test set and 9 question overall, so it might be better not to be considered at all as a separate category.

- In Sec. 5 (Discussion) and Table D1, how can MCQ scores of different benchmarks be at all comparable?

- How did the authors homogenize medical terminology in their dataset given the lack of standardization in the Arabic medical domain?

- In Appendix A, how comes the average question length is just over 8 characters?

- How was the MCQ benchmarking implemented? e.g. using logit ranking or post-processing of model answer for choice character?

---

> ### Author Response · Authors · 2025-11-23
>
> [1/n]
> Thank you for your thoughtful and constructive feedback! Below, we provide a detailed point-by-point response to each of your questions and comments.
>
> > How is it that Arabic is underrepresented in the medical domain because of its rich morphology and dialectal variation? And why the uneven linguistic landscape calls for medical LLMs? The motivation of this work in Sec.1 should be revised.
>
> We acknowledge the ambiguity of our original framing of this matter in the Introduction. As such, we have updated Section 1 in the revised manuscript as follows:
>
> *“However, it remains underrepresented in the medical domain, mainly due to limited expert‑annotated resources [1]. Arabic NLP generally presents inherent language-specific linguistic challenges [1-3], making the availability of such resources essential for assessing the performance of LLMs, especially as they are being deployed in diverse medical contexts.”*
>
> References:
> 1. Habash, Nizar Y. Introduction to Arabic natural language processing. Morgan & Claypool Publishers, 2010.
> 2. Farghaly, Ali, and Khaled Shaalan. "Arabic natural language processing: Challenges and solutions." ACM Transactions on Asian Language Information Processing (TALIP) 8.4 (2009): 1-22.
> 3. Al Moaiad, Yazeed, et al. "Challenges in natural Arabic language processing." Edelweiss Applied Science and Technology 8.6 (2024): 4700-4705.”
>
> > How were lecture notes converted to meaningful MCQs? This is an important detail that deserves to be discussed.
>
> Thank you for pointing this out. We did not use the lecture notes, so we refined this part to exclude mentioning them as it is out of scope of the study. We plan to investigate their use as part of future work, and we have added a few lines in Section 6:
>
> *“Another important area of future work is to investigate the use of Arabic-based lecture notes to advance medical Arabic NLP beyond MCQs.”*
>
> >The LLMs chosen for benchmarking do not include any Arabic-focused model. Why not? The differences in the benchmarking results could be due to the models' inherent limitation in Arabic itself rather than in medical knowledge, where bigger, proprietary models have an edge, but this angle is not explored.
>
> Thank you for your feedback on this matter. We agree with your comment and have performed additional benchmarking experiments using the following models and their respective overall accuracy scores:
>
> - General Purpose Models
>    - LLaMa-3.1-8B (17.0%)
> - Arabic-centric Models
>    - Fanar-C-1-8.7B (49.77%)
>    - Allam-7B-instruct (44.6%)
>    - Cohere c4ai-command-r7B-arabic-02-2025 (38.07%)
> - Medical Multilingual Models
>    - Medgemma-4B-it (39.02%)
>    - Apollo-7B (23.78%)
>    - Med42-8B (31.82%)
>
> The additional results provide valuable information on how Arabic-centric and medical multilingual models compare to general-purpose LLMs on MedAraBench, and highlight that even specialized models still underperform proprietary systems on this task. We have integrated these baselines into Section 4.4 and updated Table 4 (and corresponding appendix tables) to report their performance alongside the previously evaluated models. We have also updated our discussion to analyze these results in light of model size and training focus, noting that smaller or domain-specialized models do not consistently close the gap with larger proprietary models, especially on more challenging subsets.
>
> > How big were the subsets of MCQ? Guessing from Fig. E3. subset ABCDEF has only 2 questions in the test set and 9 question overall, so it might be better not to be considered at all as a separate category.
>
> There is a total of 25,694 questions in the MedAraBench dataset. The data subsets are split as follows:
> - ABCD: 19,444 questions (75.67% of the dataset)
> - ABCDE: 6,241 questions (24.29% of the dataset)
> - ABCDEF: 9 questions (0.04% of the dataset)
>
> The small size of the ABCDEF subset led to the inflation of accuracy scores in the benchmarking experiments for all models. As such, we appreciate your feedback and have omitted the ABCDEF subset as its own category in our analysis and graphs.

---

> ### Author Response · Authors · 2025-11-23
>
> [2/n]
> > In Sec. 5 (Discussion) and Table D1, how can MCQ scores of different benchmarks be at all comparable?
>
> Thank you for your feedback on this matter. We agree that MCQ scores of different benchmarks cannot be comparable as presented in Table D1, and have updated our experiments by running contemporary models on MedArabiQ, allowing for a fair comparison between contemporary and legacy models, as well as an indication on which benchmark among MedArabiQ and MedAraBench is more challenging. The results are as shown in Table 1 below:
>
> | Legacy Model         | MedArabiQ Accuracy | Contemporary Model        | MedArabiQ Accuracy | MedAraBench Accuracy |
> |----------------------|--------------------|----------------------------|-----------------------|----------|
> | claude-sonnet-3.5    | 0.535               | claude-sonnet-4-20250514   | 0.687         | 0.694 |
> | gemini-1.5-pro       | 0.575               | gemini-2.0-flash           | 0.727         | 0.654 |
> |                 |                | gpt-4.1                    | 0.808          | 0.673 |
> |  gpt-4                | 0.535               | gpt-5                      | 0.859          | 0.764 |
> |               |                   | gpt-o3                     | 0.838          | 0.765 |
> | llama-3.1-8b         | 0.262               | llama-3.3-70b-instruct     | 0.495         | 0.547 |
> | qwen-2.5-7b          | 0.380                 | qwen-plus                  | 0.657          | 0.618  |
>
>
> Our experimental results on the MedArabiQ benchmark show that all contemporary models outperform legacy models on the MCQ task. Additionally, our results show that gemini-2.0-flash, gpt-4.1, gpt-5, gpt-o3, and qwen-plus perform better on the MedArabiQ benchmark, while claude-sonnet-4-20250514 and llama-3.3-70b-instruct perform better on MedAraBench. This may indicate that the MedAraBench benchmark is more challenging overall, and thus provides larger value for the advancement of medical Arabic NLP.
>
> It is important to note that legacy models gemini-1.5-pro and claude-3.5-sonnet were deprecated, which prevented us from evaluating them on MedAraBench to provide a better comparison between benchmarks.
>
> We have updated Table D1 in the manuscript to better reflect the results of our new experiments.
>
> > How did the authors homogenize medical terminology in their dataset given the lack of standardization in the Arabic medical domain?
>
> We recognize that Arabic medical terminology standardization is an ongoing field-wide challenge, and acknowledge that this may not be a major issue for benchmarking LLMs since real-world medical QA may not necessarily be standardized or neat. Hence, in the scope of MedAraBench, we did not perform any external standardization of medical terminology.
>
> We further elaborate on this matter in Section 3.1 (lines 194-204):
>
> *“It is important to note that the homogeneity regarding medical terminology consistency is essential in assessing the capabilities of LLMs on standardized Arabic medical tasks. As it pertains to MedAraBench, we did not perform any external standardization across the dataset as we acknowledge that this may not be a major issue from a benchmarking perspective since real-world medical QA may not necessarily be standardized in terms of terminology. *
> *We acknowledge that Arabic medical terminology standardization remains an ongoing challenge, with unified vocabulary frameworks still under development across Arabic-speaking regions [1][2]. The expert clinician evaluations described in Section 3.2 serve as quality control, with reviewers achieving high agreement on all metrics despite the absence of standardization.”*
>
> This is due to the fact that the dataset reflects real-world Arabic medical education as it exists, not as an idealized/ standardized version, which would provide us valuable insights into LLM responses when they encounter this variability in actual clinical settings. Furthermore, despite not imposing any external standardization, we ensure the quality of our data through both board-certified physician evaluations, which would have surfaced any inconsistencies if present.
>
> References:
>
> 1. Y. M. Attia, “The Challenges of Achieving Dynamic Equivalence in the Arabic Translation of Medical Terminology for Heart and Brain Diseases: Difficulties and Recommendations,” مجلة کلية الآداب .جامعة بورسعيد, vol. 30, no. 30, pp. 14–31, Oct. 2024, doi: https://doi.org/10.21608/jfpsu.2024.307825.1371
> 2. Mohammed and A. Maria, “An Analysis of the Inconsistency Problem in the Translation of Medical Terms from English into Arabic.,” Univ-ouargla.dz, 2023, https://dspace.univ-ouargla.dz/jspui/handle/123456789/33502.

---

> > ### Author Response · Authors · 2025-11-23
> >
> > [3/3]
> > > In Appendix A, how comes the average question length is just over 8 characters?
> >
> > Thank you for catching this error. This was an error from our end while calculating the average number of characters per question and generating the graphs. We reran the analysis and found that the average question length is 37.86 characters, while the overall answer length (options A-D altogether) is 171.09 characters, which makes more sense. We have updated the values and graphs in Appendix A accordingly.
> >
> > > How was the MCQ benchmarking implemented? e.g. using logit ranking or post-processing of model answer for choice character?
> >
> > We implemented MCQ benchmarking using answer extraction via post-processing. Models were prompted to output the answer choice letter (A-D) directly, and we parsed their text responses using pattern matching to extract the answer. We have updated Section 3.3 in the manuscript to properly reflect our benchmarking protocol.
> >
> > > The reference to Appendix A is missing in Sec. 3.1
> >
> > > MedArabiQ is missing from Table 1
> >
> > >  Line 331 (Sec 4.3) is truncated.
> >
> > Thank you for catching these errors. We added the missing reference to Appendix A in Section 3.1, added MedArabiQ to Table 1, and fixed the truncation on Line 331.
> > We kindly ask you to evaluate our submission in light of the above feedback and improvements made to our manuscript. Thank you for your time and effort.

---

> > ### Author Response · Authors · 2025-11-27
> >
> > Dear Reviewer,
> >
> > This is a brief note to kindly invite you to consult our rebuttal and revised manuscript, where we address the points raised in your review in detail. Please let us know if any clarification would be helpful.
> >
> > Thank you again for your time and consideration.

---

### Author Response · Authors · 2025-11-23

Dear AC/ SACs,

We sincerely thank the reviewers for taking the time to provide thoughtful feedback, comments, and questions, as well as recognizing the strengths of our work. We would like to highlight the positive feedback we received from the reviewers:
- MedAraBench was recognized as a useful resource for LLM benchmarking and Arabic language understanding.
- MedAraBench’s main strength is that it spans 19 specialties and various difficulty levels, supporting fine-grained evaluation of LLM performance and filling a critical resource gap in healthcare AI research.
- Reviewers appreciated our significant effort in manually digitizing and expertly validating data from non-digital academic resources
- Inclusion of a human expert evaluation component, which is commendable and adds qualitative depth to the study.
- Contributes to Arabic NLP, a domain with limited existing benchmarks and resources

Additionally, we believe we have thoroughly addressed the reviewers’ concerns offering new results and clarifications:
- We clarified our work’s motivation and the potential of developing high-quality LLMs in advancing Arabic and medical NLP and providing care for patients around the globe.
- We clarified the data collection process and how each question was assigned its medical specialty according to its source material.
- We benchmarked 7 additional models according to the following criteria to allow for a fair evaluation of specialized Arabic and medical models:
  - General Purpose Models
       - LLaMa-3.1-8B
  - Arabic-centric Models
       - Fanar-C-1-8.7B
       - Allam-7B-instruct
       - Cohere c4ai-command-r7B-arabic-02-2025
  - Medical Multilingual Models
       - Medgemma-4B-it
       - Apollo-7B
       - Med42-8B
- We updated our LLM-as-a-judge framework to follow a binary scale using the top-performing models on the MedAraBench dataset. We also conducted statistical tests to evaluate model alignment with expert evaluations.
- We compared the performance of contemporary vs legacy models. This experiment also allowed us to compare the difficulty of the existing benchmark and our MedAraBench, thus providing insights on its value.
- We clarified model grouping and resources, and compared training-data sizes for open- and closed-source models.
- We performed few-shot and LoRA finetuning experiments on Llama-3.1-8B-instruct, allowing us to assess the impact of MedAraBench data on open-source model performance

We kindly ask you to evaluate our submission in light of the above feedback and improvements made to our manuscript. We believe we have addressed all major concerns raised by the reviewers, which has significantly improved the clarity and reliability of our manuscript.


Thank you for your time and your valuable service to the research community.

---

### Meta-Review · Area_Chair_ax3f · 2026-01-01

**Summary:**

The submission presents MedAraBench, a large-scale Arabic medical Question Answering (QA) dataset comprising approximately 24,000 multiple-choice questions across 19 medical specialties and five difficulty levels. The authors constructed the dataset through manual digitization of academic materials from the Arabic-speaking region. The work includes a comprehensive evaluation using both expert human annotators and an LLM-as-a-judge framework, alongside benchmarking of various state-of-the-art proprietary and open-source Large Language Models (LLMs).

The decision to accept is based on the significant contribution this dataset represents for the low-resource field of Arabic Medical NLP. The authors have demonstrated a high degree of responsiveness during the rebuttal phase, conducting extensive new experiments to address methodological concerns regarding baselines, evaluation metrics, and comparative analysis.

**Reviewer Concerns:**

Addressed Concerns:
- Missing Baselines: Reviewers m6wR and BRmF noted the absence of relevant Arabic-centric and medical-specific models. The authors successfully integrated benchmarks for BiMediX, Med42, Apollo, Allam, Fanar, and Medgemma into the revised manuscript.
- LLM-as-a-Judge Methodology: Reviewer BRmF heavily criticized the initial use of a Likert scale for LLMs versus a binary scale for humans, and the lack of justification for the judge selection. The authors revised the entire setup to use a binary scale, calculated Pearson correlations to validate GPT-03 as the most aligned judge, and updated the results accordingly.
- Cross-Benchmark Comparisons: Reviewers noted that comparing new model scores on MedAraBench against old model scores on MedArabiQ was invalid. The authors re-ran contemporary models on the MedArabiQ benchmark, enabling a valid direct comparison which demonstrated that MedAraBench is indeed more challenging.
- Dataset Anomalies: Reviewer BRmF flagged suspicious perfect accuracy on the "ABCDEF" subset. The authors clarified this subset contained only 9 questions and removed it from the analysis to prevent skewed metrics.
- Evaluation Scope: Reviewer m6wR suggested validating the training set. The authors performed LLM-as-a-judge evaluation on the full training set and added few-shot and LoRA fine-tuning experiments to demonstrate the training set's utility.
- Scale Comparisons: To address concerns about comparing open vs. closed models unfairly, the authors reorganized the results by model category (General, Arabic-centric, Medical) and explicitly listed parameter counts/training data sizes.

Outstanding Concerns:
- Dataset Imbalance: The dataset remains skewed toward Year-1 difficulty questions (approx. 56%). The authors acknowledged this is a limitation of the available source material but provided analysis showing that models still struggle with the harder subsets (Y3), validating the benchmark's utility.
- Terminology Standardization: The dataset lacks strict terminology standardization. The authors argued this was a design choice to reflect real-world clinical/educational variation in the region, which is a reasonable justification for a benchmark dataset.

**Reviewer Scores:**

- Reviewer ratc (Current Score: 8):
    - This reviewer was already positive. The authors addressed their minor concerns regarding motivation, formatting, and the exclusion of the "ABCDEF" subset.
    - Projected Score: 8 (Accept). The rebuttal reinforced the paper's quality, confirming the reviewer's initial positive assessment.

- Reviewer m6wR (Current Score: 4):
    - This reviewer's main hesitancy stemmed from missing baselines (specifically BiMediX and Apollo) and the lack of training set validation.
    - Projected Score: 6 (Weak Accept) or 7 (Accept). The authors explicitly ran the requested baselines and performed the suggested training set validation. Since the "Fair" rating for soundness was based on these missing elements, the score would almost certainly improve.

- Reviewer BRmF (Current Score: 2):
    - This reviewer provided the most critical feedback, focusing on methodological flaws in the LLM-judge setup (scale mismatch), invalid comparisons, and missing comparisons.
    - Projected Score: 5 (Borderline Accept) or 6 (Weak Accept). The authors went to great lengths to "fix" the paper based on this review. They switched the evaluation scale to binary (directly addressing the validity concern), re-ran the cross-benchmark analysis (addressing the comparison concern), and added the requested models. While the reviewer might still view the contribution as "incremental" (dataset vs. new method), the methodological grounds for rejection have been effectively removed.

---

### Decision · Program_Chairs · 2026-01-26

Accept (Poster)